# Centriolar satellites are required for efficient ciliogenesis and ciliary content regulation

Ezgi Odabasi[1], Seref Gul[1,2], Ibrahim H Kavakli[1,2] & Elif N Firat-Karalar[1,*] ID

## Abstract

Centriolar satellites are ubiquitous in vertebrate cells. They have recently emerged as key regulators of centrosome/cilium biogenesis, and their mutations are linked to ciliopathies. However, their precise functions and mechanisms of action remain poorly understood. Here, we generated a kidney epithelial cell line (IMCD3) lacking satellites by CRISPR/Cas9-mediated PCM1 deletion and investigated the cellular and molecular consequences of satellite loss. Cells lacking satellites still formed full-length cilia but at significantly lower numbers, with changes in the centrosomal and cellular levels of key ciliogenesis factors. Using these cells, we identified new ciliary functions of satellites such as regulation of ciliary content, Hedgehog signaling, and epithelial cell organization in three-dimensional cultures. However, other functions of satellites, namely proliferation, cell cycle progression, and centriole duplication, were unaffected in these cells. Quantitative transcriptomic and proteomic profiling revealed that loss of satellites affects transcription scarcely, but significantly alters the proteome. Importantly, the centrosome proteome mostly remains unaltered in the cells lacking satellites. Together, our findings identify centriolar satellites as regulators of efficient cilium assembly and function and provide insight into disease mechanisms of ciliopathies.

**Keywords** centriolar satellites; Hedgehog; PCM1; primary cilium
**Subject Categories** Cell Adhesion, Polarity & Cytoskeleton

## Introduction

The vertebrate centrosome/cilium complex is composed of centrosomes, cilia, and centriolar satellites. The evolutionarily conserved microtubule-based cylindrical structures, centrioles, recruit the pericentriolar material to form the centrosome [1,2]. In quiescent and differentiated cells, the older mother centriole functions as a basal body to nucleate the formation of flagella and cilia, microtubule-based structures projecting out from the apical surface of cells. In addition to these conserved structures, specific to vertebrate cells is the array of 70- to 100-nm membraneless granules termed centriolar satellites, which localize to and move around the centrosome/cilium complex in a microtubule- and dynein-/dynactin-dependent manner and are implicated in active transport or sequestration of centrosome and cilium proteins [3–5].

The assembly and function of the centrosome/cilium complex are regulated in response to cell cycle cues and environmental factors. In interphase cells, the centrosome nucleates and organizes the microtubule array, which functions in vesicular trafficking, cell motility, and maintaining cell shape and polarity [6,7]. Centrosomes duplicate in S phase and form the bipolar mitotic spindle during mitosis [8,9]. In quiescent cells, centrioles dock to the plasma membrane to form the primary cilia, which function as a nexus for developmentally important signaling pathways like Hedgehog and Wnt signaling [10]. Primary cilium formation is a highly complex and regulated process that is mediated through intracellular pathway in the majority of cells and extracellular pathway in polarized epithelial cells [11,12]. Given that centrosomes and cilia are indispensable for key cellular processes, their structural and functional defects are associated with a variety of human diseases including cancer and ciliopathies [8,13]. Elucidating the mechanisms that regulate the assembly and function of the centrosome/cilium complex in time and space is required to define the molecular defects underlying these diseases.

Centriolar satellites have recently emerged as key regulators of centrosome/cilium complex biogenesis [3,5,14]. Consistently, mutations affecting satellites components are linked to diseases associated with defects in the centrosome/cilium complex such as ciliopathies, primary microcephaly, and schizophrenia [3,15–17]. Ciliopathies are genetic diseases characterized by a multitude of symptoms including retinal degeneration and polycystic kidney disease [18,19]. Why defects of the broadly expressed centrosome/cilium complex components are reflected as clinically restricted and heterogeneous phenotypes among different tissue types is unknown. Corroborating the link between satellites and ciliopathies, loss of satellites in zebrafish caused characteristics of cilium dysfunction, analogous to those observed in human ciliopathies [20].

Although centriolar satellites are ubiquitous in vertebrate cells, their size and number vary among different cell types, and change in response to signals including cell cycle cues and stress [21–23]. In different cell types and tissues, their cellular distribution ranges from clustering at the centrosomes, nuclear envelope, and/or basal

1   Department of Molecular Biology and Genetics, Koç University, Istanbul, Turkey
2   Department of Chemical and Biological Engineering, Koç University, Istanbul, Turkey
    *Corresponding author. Tel: +90 212 3381677; Fax: +90 212 3381559; E-mail: ekaralar@ku.edu.tr

bodies, to scattering throughout the cytoplasm [24–26]. This variation suggests that satellites might have cell type- and tissue-specific functions. Elucidation of these functions could provide important insight into the mechanisms underlying the phenotypic heterogeneity of ciliopathies.

Over 100 proteins have been defined as satellite components through their co-localization with or proximity to PCM1 (pericentriolar material-1), the scaffolding protein of satellites that mediates their assembly and maintenance [3,27]. Phenotypic characterization of various satellite proteins has defined functions in cilium formation, centrosome duplication, microtubule organization, mitotic spindle formation, chromosome segregation, actin filament nucleation and organization, stress response, and autophagy [3,23,28–36]. However, these may not be satellite-specific functions per se, because all satellite proteins identified so far except for PCM1 localize to both satellites and centrosomes and/or cilia.

Satellite-specific functions have been identified through transient or constitutive depletion of PCM1 from cells, which causes satellite disassembly [36,37]. Transient PCM1 depletion causes defects in protein targeting to the centrosome, interphase microtubule network organization, cell cycle progression, cell proliferation, and cilium formation, as well as neuronal progenitor cell proliferation and migration during cortical development [37–41]. A recent study generated human PCM1$^{-/-}$ retinal pigmental epithelial (RPE1) cells and showed that satellites are essential for cilium assembly, where they sequester the E3 ubiquitin ligase Mib1 and prevent degradation of the key ciliogenesis factor Talpid3, which is required for the recruitment of ciliary vesicles [36]. However, because RPE1::PCM1$^{-/-}$ cells did not ciliate, they did not allow assessment of satellite-specific functions in ciliary signaling and ciliary targeting of proteins, which are among the predominant phenotypic defects underlying ciliopathies. Therefore, a major unresolved question that pertains to our understanding of satellites and their relationship to specific ciliopathies is the identification of the full repertoire of satellite-specific functions.

In this study, we generated kidney epithelial cells that are null for centriolar satellites by ablating the core satellite component PCM1 using genome editing and studied the cellular and molecular consequences specifically of satellite loss on both centrosome- and cilium-related cellular processes. We chose mouse inner medullary collecting duct (IMCD3) cells to complement RPE1 cells because they are Hedgehog-responsive, they form epithelial spheroids that are dependent on cilia functions when grown in a three-dimensional matrix, and they ciliate using the extracellular ciliogenesis pathway, whereas RPE1 cells use the intracellular pathway. Together, our results identify satellites as key regulators of ciliogenesis, ciliary signaling, tissue architecture, protein targeting, and cellular proteostasis, and provide insight into the mechanisms underlying specific ciliopathies.

# Results

### Disruption of *PCM1* causes loss of centriolar satellites in kidney epithelial cells

To determine the cellular functions of centriolar satellites, we generated satellite-less cells by disrupting the *PCM1* gene in mouse kidney epithelial IMCD3 cells. Homozygous null mutations in both alleles of the *PCM1* locus were made using CRISPR/Cas9-mediated genome editing with guides designed to target exon 3 (protein-coding exon 2) in IMCD3 cells (Fig EV1A and B). We isolated three PCM1$^{-/-}$ IMCD3 clones (hereafter IMCD3 PCM1 KO) and one control colony (hereafter WT) that was transfected with the plasmid encoding the scrambled gRNA. Sequencing of the PCM1 alleles identified these clones as compound heterozygotes bearing premature stop codons that result from small deletions of < 20 base pairs and/or insertion of one or two base pairs around the cut site (Fig EV1A and B). Immunoblot analysis of whole-cell lysates with two different polyclonal antibodies, one directed against the N-terminal 1–254 amino acids and the other against the C-terminal 630–726 amino acids of PCM1, showed that PCM1 was not expressed in the IMCD3 PCM1 KO clones and that LAP-PCM1 was expressed in the rescue line (Fig 1A). Immunofluorescence analysis of these clones with the N-terminal antibody and an antibody targeting the C-terminal 1,665–2,026 amino acids of PCM1 further validated lack of PCM1 expression (Figs 1B and EV1C). The absence of PCM1 signal in the PCM1 KO clones with the C-terminal PCM1 antibody eliminated the possibility that in-frame gene products downstream of the gRNA-target site were initiated, and showed that PCM1 alleles in these clones are likely to be null mutations, which was confirmed by mass spectrometry-based quantitative global proteome analysis described below.

To confirm that PCM1 KO cells lack satellite structures, we determined the localization of other known satellite components in these cells. While Cep131 and Cep72 localized to both the centrosome and satellites in control cells, their localization was restricted to the centrosomes in IMCD3 PCM1 KO cells, and there were no granules around the centrosome that were positive for satellite markers in these cells (Fig 1C). Stable expression of LAP-tagged human full-length PCM1 and PCM1 (1–1,200) in IMCD3 PCM1 KO clone#1 rescued satellite assembly and localization defects, confirming the specificity of these phenotypes (Fig 1D). Of note, even though PCM1 (1–1,200) expression formed PCM1-positive granules that recruited other satellite components including Cep131, the size of the granules was larger than the ones formed through expression of full-length PCM1 (Fig 1D), suggesting a regulatory role for the C-terminal part of PCM1 in satellite assembly. These results show that IMCD3 PCM1 KO clones are devoid of satellites and that PCM1 is essential for the assembly of satellites and sequestration of centrosome proteins to the satellites.

### Satellites are required for efficient ciliogenesis, but not for cell proliferation, cell cycle progression, and centriole duplication

Many satellite proteins, including PCM1, have previously been implicated in ciliogenesis-related functions [36,39,40,42]. While a recent PCM1 knockout study in RPE1 cells and our characterization of RPE1 PCM1 KO cells generated by genome editing identified essential roles for satellites in cilium assembly [36] (Fig EV3A–D), the function of satellites in a wide range of centrosome-/cilium-related cellular processes has not been studied in clean genetic backgrounds. To address this, IMCD3 PCM1 KO cells were serum-starved for 24 h and 48 h and the percentage of ciliated cells were quantified by staining cells for Arl13b, a marker for the ciliary membrane, and acetylated tubulin, a marker for the ciliary axoneme. Lack of

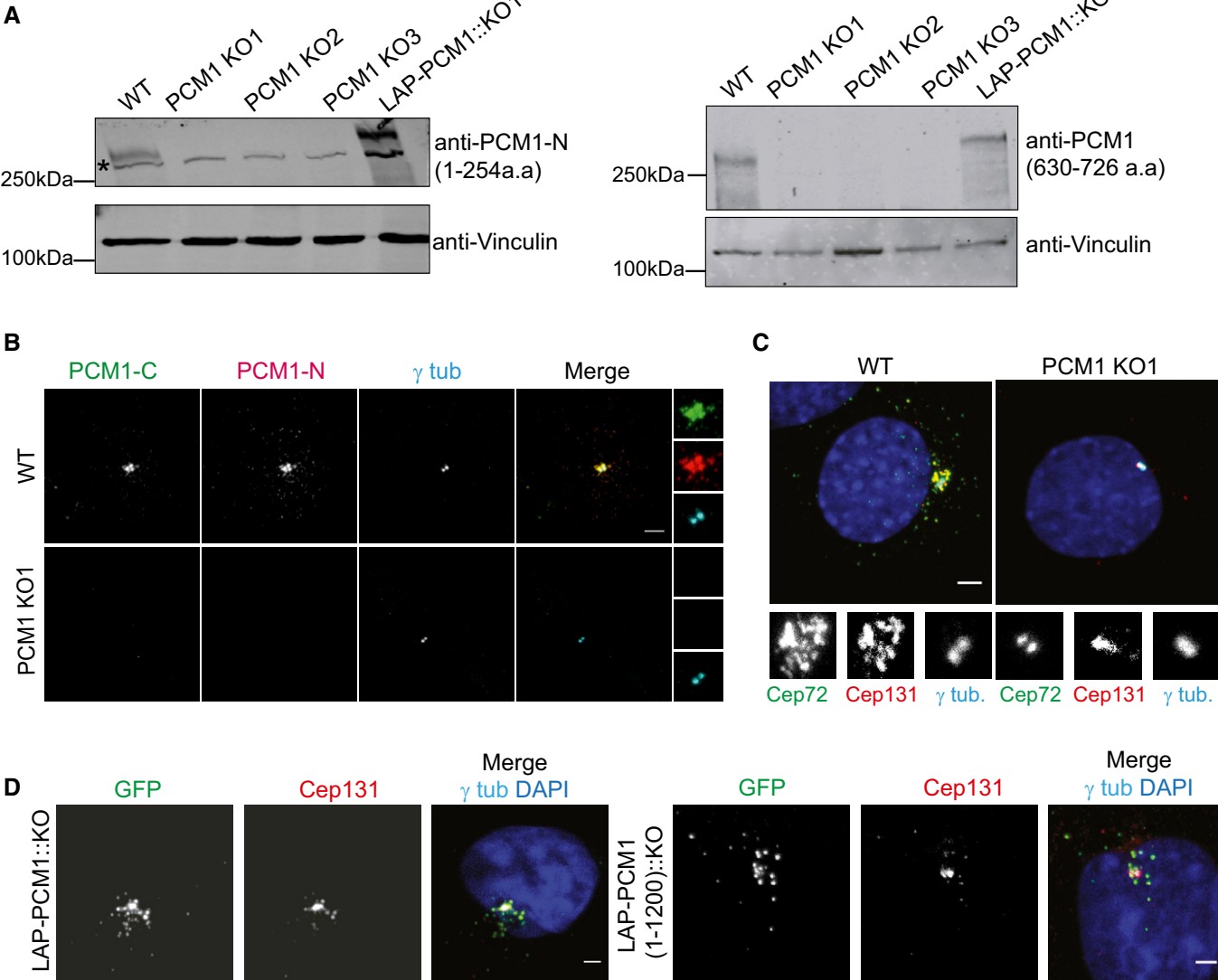

**Figure 1. IMCD3 PCM1 KO cells are devoid of satellite structures.**

A, B IMCD3 PCM1 KO cells do not express PCM1. (A) Immunoblot analysis of whole-cell lysates from control cells, three IMCD3 PCM1 KO lines, and one IMCD3 PCM1 KO::LAP-PCM1 rescue line with two different PCM1 antibodies, one raised against the N-terminal amino acids 1–254 and the other against amino acids 630–726. (B) Immunofluorescence analysis of control and IMCD3 PCM1 KO1 cells. Cells were fixed and stained for centrosomes with anti-γ-tubulin antibody and PCM1 with PCM1-N antibody raised against its N-terminal (amino acids 1–254) fragment and PCM1-C antibody raised against its C-terminal (amino acids 1,665–2,026) fragment. Scale bar, 4 μm.

C Localization of satellite proteins Cep72 and Cep131 is restricted to the centrosome in IMCD3 PCM1 KO cells. Cells were fixed and stained for Cep72, Cep131, and gamma-tubulin. DNA was stained with DAPI. Scale bar, 5 μm.

D Stable expression of LAP-PCM1 and LAP-PCM1 (1–1,200) rescues the satellite assembly defects in IMCD3 PCM1 KO cells. Cells were fixed and stained with GFP, Cep131, and gamma-tubulin antibodies. DNA was stained with DAPI. Scale bar, 4 μm

satellites resulted in a significant decrease in the fraction of ciliated cells in serum-starved IMCD3 cells relative to control cells serum-starved for 24 and 48 h (Fig 2A and B). This is in contrast to RPE1 PCM1 KO cells, which did not ciliate [36]. All three IMCD3 PCM1 KO clones had similar ciliogenesis defects (Fig 2B), and we used clone # 1 for subsequent rescue and phenotypic characterization experiments (hereafter IMCD3 PCM1 KO). The ciliogenesis phenotype in IMCD3 PCM1 KO cells was rescued by stable expression of LAP-tagged full-length human PCM1, confirming that the

ciliogenesis phenotype is a specific consequence of satellite loss (Fig 2A and B). Despite the reduction in the ciliating population in IMCD3 PCM1 KO cells, the cilia that formed in these cells were normal in length (wt: 3.14 μm ± 0.8, KO: 3.38 μm ± 1.18), which suggests that PCM1 acts during cilia initiation (Fig 2C).

We next asked whether the ciliogenesis defects of IMCD3 PCM1 KO cells are an indirect consequence of defects in cell cycle progression or centriole duplication. We tested the cell cycle-related phenotypes using a combination of assays. First, we performed

proliferation assays with control and IMCD3 PCM1 KO cells by plating equal number of cells and counting them at indicated time intervals, which showed that loss of satellites did not have a significant impact on cell doubling times (Fig 3A). Second, we analyzed the fraction of control and IMCD3 PCM1 KO cells in different phases of the cell cycle using flow cytometry and showed that both had similar cell cycle profiles (Figs 3B and EV1D). Moreover, we performed live imaging of control and IMCD3 PCM1 KO cells stably expressing mCherry-H2B to assay for cell cycle progression defects. Both control and PCM1 KO cells had similar mitotic times (WT: 29 ± 7.7 min, KO: 28.1 ± 6.7 min; Fig 3C and D). Finally, we quantified centriole number in asynchronous control and IMCD3 PCM1 KO cells by centrin2 and centrin3 labeling and showed that PCM1 KO cells had similar centriole counts relative to control cells (Fig 3E). To investigate the generality of these phenotypes, we generated two RPE1 PCM1 KO clones using genome editing and confirmed their satellite null state by immunofluorescence, immunoblotting, and sequencing (Fig EV2A–D). Like IMCD3 PCM1 KO cells, RPE1 PCM1 KO cells did not have defects in cell proliferation, distribution of cell cycle phases, and centriole duplication (Fig EV2E–H). Collectively, these data indicate that satellites are not required for cell cycle progression and centriole duplication, and that the reduced ciliogenesis phenotype of PCM1 KO cells is not caused by defects in these processes.

## Satellites have variable effects on regulating centrosomal and cellular abundance of key ciliogenesis factors

Satellites regulate centrosomal recruitment of proteins including centrin, pericentrin, and ninein among others [37]. We

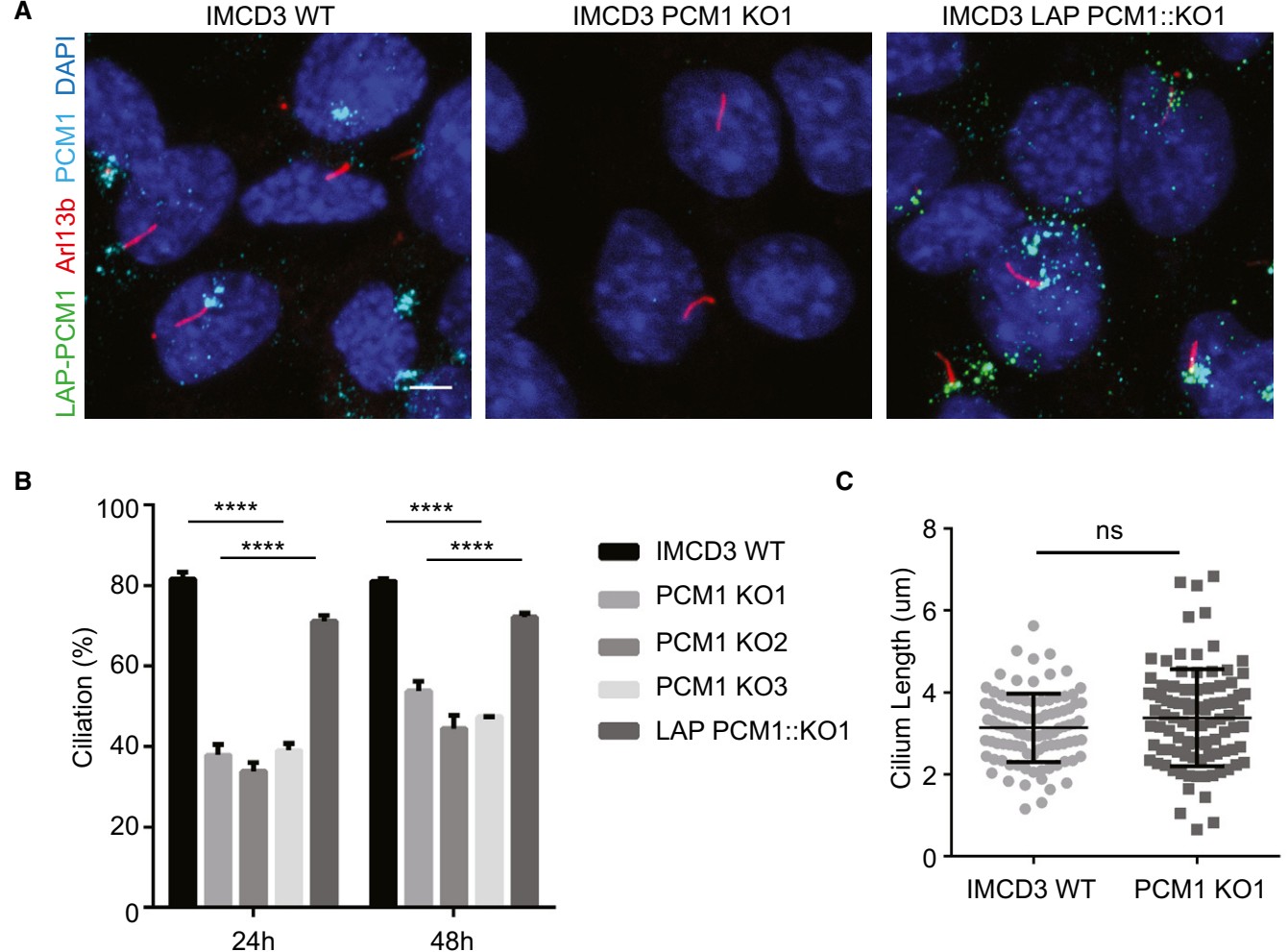

**Figure 2. Satellites are required for efficient ciliogenesis.**

A Effect of satellite loss on cilium formation. Control cells, IMCD3 PCM1 KO cells and LAP-PCM1-expressing IMCD3 PCM1 KO cells serum-starved for the indicated times, and percentage of ciliated cells was determined by staining for acetylated tubulin, Arl13b, and DAPI. Scale bar, 4 μm.

B Quantification of ciliogenesis and rescue experiments. Results shown are the mean of three independent experiments ± SD (500 cells/experiment, ****P < 0.0001, *t*-test).

C Effect of satellite loss on cilium length. IMCD3 cells serum-starved for 24 h, and cilium length was determined by staining for acetylated tubulin and DAPI. Quantification of cilium length was done from three independent experiments with an average of 90 cilia measured per experiment. Error bars represent SD. Horizontal line represents the mean value of each group. There is no significant difference between control and IMCD3 PCM1 KO cells (*t*-test).

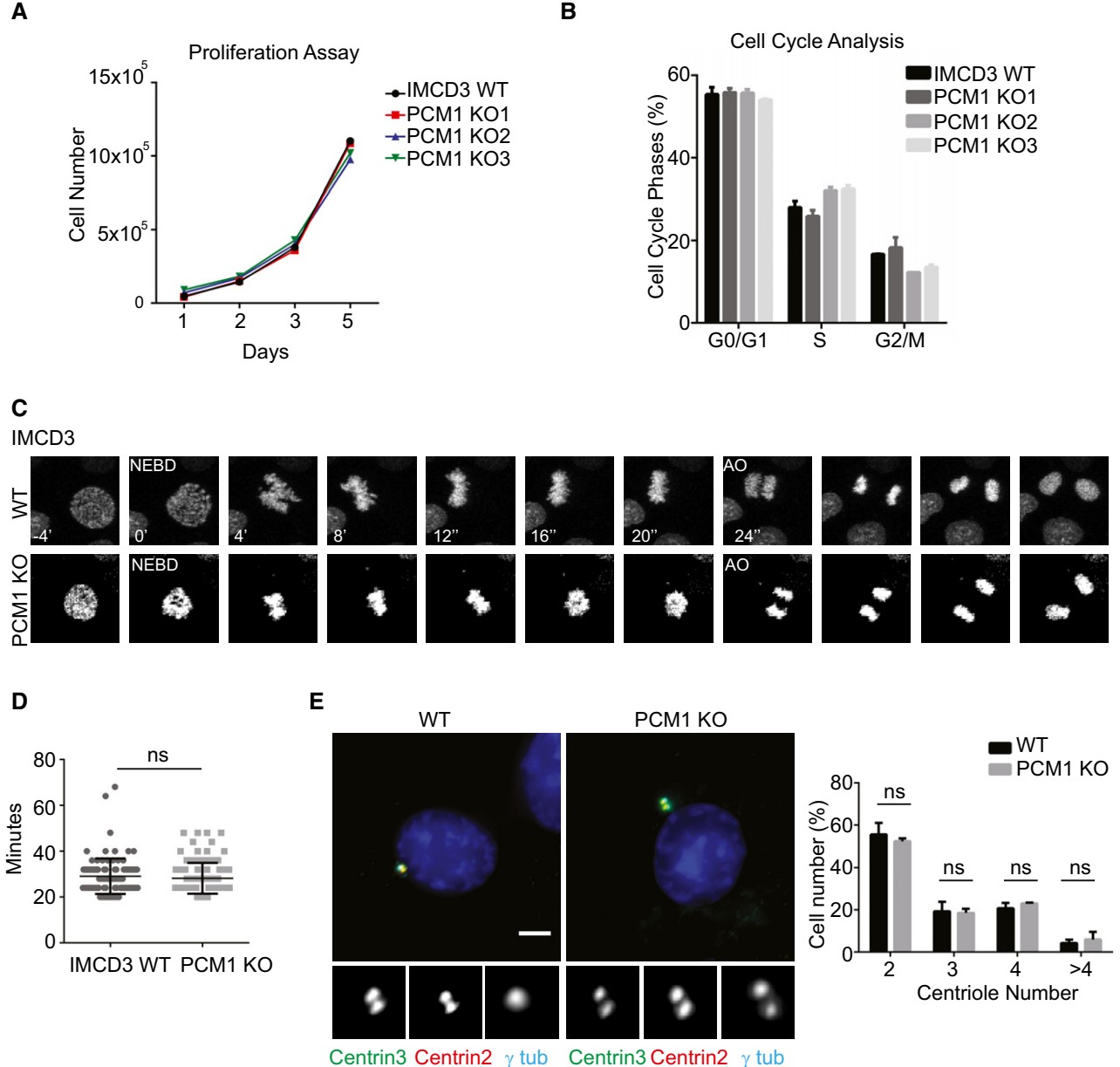

**Figure 3. Satellites are not required for cell proliferation, cell cycle progression, and centriole duplication.**

A  Effects of satellite loss on cell proliferation. $2 \times 10^5$ cells were plated and counted at 1, 2, 3, and 5 days. Data points show mean $\pm$ SD of three independent experiments. There was no significant difference between control and IMCD3 PCM1 KO cells at any time point.

B  Effect of satellite loss on percentage of cells in different cell cycle phases. Flow cytometry analysis of control or IMCD3 PCM1 KO cells. Data points show mean $\pm$ SD of three independent experiments.

C  Effect of satellites loss on mitotic times. Control and IMCD3 PCM1 KO cell stably expressing mCherry-H2B were imaged every 4 min for 24 h. Representative still frames from time-lapse experiments are shown.

D  Mitotic time was quantified as the time interval from nuclear envelope breakdown (NEBD) to anaphase onset. Data points show mean $\pm$ SD of two independent experiments. Control and IMCD3 PCM1 KO cells had similar mitotic times. $t$-test was used for statistical analysis.

E  Effect of satellite loss on centriole duplication. Asynchronous control and IMCD3 PCM1 KO cells were stained for centrioles using centrin2 and centrin3 antibodies, centrosomes using gamma-tubulin antibody, and DNA with DAPI stain. Quantification results shown are the mean of two independent experiments $\pm$ SD (100 cells/experiment, $t$-test, ns: non-significant). Scale bar, 4 μm.

hypothesized that the ciliogenesis defects of IMCD3 cells might be due to inefficient targeting of key ciliogenesis factors to the centrosome. To test this, we used quantitative immunofluorescence to determine the centrosomal levels of proteins that are components of distal appendages, transition zone proteins, IFT machinery as well as activators and suppressors of ciliogenesis in asynchronous and

serum-starved control cells, IMCD3 PCM1 KO cells, and the LAP-PCM1 rescue clone.

Distal appendages function in docking of the mother centriole to the membrane during ciliogenesis, and transition zone regulates the entry and exit of ciliary cargo [43,44]. The defects in ciliogenesis were not because of mislocalization of distal appendage or the transition zone proteins since Cep164 and Cep290 localized correctly to the centrosomes in both control and IMCD3 PCM1 KO cells (Figs 4A and B, and EV3A and B). Of note, the centrosomal levels of Cep164 were higher in PCM1 KO cells, which were rescued by stable LAP-PCM1 expression (Figs 4A and EV3A). This increase might be due to the increased abundance of Cep164 in total cell lysates in PCM1 KO cells (Fig 4G). CP110 localizes to the distal ends of centrioles and is removed from the mother centrioles during ciliogenesis [45]. The centrosomal levels of CP110 were similar in control and IMCD3 PCM1 KO cells (Figs 4C and EV3C). Importantly, IMCD3 PCM1 KO cells had a significant reduction in the basal body levels of IFT-B protein Ift88 in both asynchronous and serum-starved cells (Figs 4D and EV3D), which was rescued by stable expression of LAP-PCM1 (Figs 4D and EV3D). Given that the IFT-B machinery is required for the assembly of the ciliary axoneme [46–49], satellites might promote cilium assembly at a step upstream to the recruitment of the IFT machinery at the base of cilia.

The inhibition of ciliogenesis in RPE1 PCM1 KO cells was shown to be a consequence of increased cellular and centrosomal levels of the E3 ubiquitin ligase Mib1, which led to a reduction in the centrosomal levels of the key ciliogenesis factor Talpid3/KIAA0586 through its destabilization [36]. We investigated whether loss of satellites had similar molecular consequences in IMCD3 cells. Despite the significant $1.5 \pm 0.26$-fold increase in the cellular abundance of Mib1 in IMCD3 PCM1 KO cells (Fig EV3G), its centrosomal levels did not change in these cells relative to control cells (Figs 4E and EV3E). The centrosomal and cellular levels of Talpid3 in IMCD3 PCM1 KO cells were higher than those in control cells, which were rescued by stable LAP-PCM1 expression (Figs 4F and G, and EV3F). Thus, in contrast to RPE1 cells, Mib1 sequestration and consequent Talpid3 degradation at the centrosome were not induced in satellite-less IMCD3 cells despite their increased cellular levels. Collectively, these results identify differences in the efficiency of the centrosomal targeting of key ciliogenesis factors as a likely mechanism for the phenotypic variability of satellite-less IMCD3 and RPE1 cells during cilium formation.

We also performed immunoblot analysis of whole-cell lysates of asynchronous and serum-starved control and IMCD3 PCM1 KO cells to determine whether the cellular abundance of these proteins was affected. While there was an overall increase in the abundance of Mib1, Cep164, Cep131, and Ift88 and overall decrease in SLAIN2, the abundance of Bbs4, SSXIIP, Cep290, Cp110, Talpid3, and Arl13b was unaffected (Figs 4G and EV3G). Thus, loss of satellites has differential effects on the abundance of different centrosome and cilium proteins. Notably, except for Cep164, changes in abundance of these proteins were not reflected by similar changes in the recruitment of these proteins to the centrosome. For example, while Ift88 cellular levels increase significantly, its centrosomal levels had a significant reduction.

## Satellites regulate ciliary protein content

Analogous to their function in regulating protein targeting to the centrosome, we hypothesized that satellites might also regulate ciliary recruitment of proteins. To test this, we examined the composition of the ciliary shaft and the ciliary membrane of control and IMCD3 PCM1 KO cells by determining the ciliary levels of various proteins using quantitative immunofluorescence. Control and IMCD3 PCM1 KO cells had similar levels of ciliary Arl13b, which associates with the ciliary membrane via palmitoylation [50] (Fig 5A). However, IMCD3 PCM1 KO cells had significantly reduced ciliary levels of the IFT-B protein Ift88, somatostatin receptor 3 (SSTR3-LAP), and the serotonin 6 receptor (HTR6-LAP) (Fig 5B–D). The decrease in ciliary Ift88 in IMCD3 PCM1 KO cells was rescued by stable expression of LAP-PCM1 (Fig 5D). To investigate the function of satellites in the dynamic recruitment of ciliary proteins, we performed half- and full-cilium fluorescence recovery after photobleaching (FRAP) to measure the dynamic behavior of HTR6-LAP and SSTR6-LAP in IMCD3 cells [51]. Both proteins did not recover in full-cilium FRAP experiments in control and PCM1 KO cells (Fig EV4A). Likewise, the half-time (LAP-SSTR3 in WT: $17.64 \pm 3.41$ s, PCM1 KO: $19.96 \pm 4.31$ s; LAP-HTR6 in WT: $43.24 \pm 17.45$ s, PCM1 KO: $53.87 \pm 20.15$ s) and percentage of their recovery (SSTR3-LAP in WT: $40.19 \pm 6.73$, in PCM1 KO: $35.85 \pm 7.64$ s; LAP HTR6 in WT: $45.36 \pm 13.92$, PCM1 KO: $42.57 \pm 5.94$) in half-cilium FRAP experiments were similar (Fig EV4B–D).

The ciliary localization of some Hedgehog pathway components is dynamically regulated in response to pathway activation. To elucidate the function of satellites in this dynamic regulation and to investigate whether the cilia assembled in IMCD3 satellite-less cells can still respond to Hedgehog ligands, we quantified the ciliary recruitment of the seven-transmembrane protein Smo, which moves to cilia in response to Sonic Hedgehog (Shh) ligands [52]. To this end, we stimulated control and IMCD3 PCM1 KO cells with 200 nM Smoothened agonist (SAG) in a time course manner and quantified the percentage of cilia with Smo. A significant ($P < 0.01$) twofold reduction in the percentage of cilia with Smo was observed in PCM1 KO cells ($40.1\% \pm 7.9$ of total) relative to control cells ($80.6\% \pm 6.9$ of total) at 4 h, and $45.9\% \pm 18.2$ of total in PCM1 KO cells relative to $75.5\% \pm 12.5$ of total in control cells at 8 h (Fig 6A). Notably, control and IMCD3 PCM1 KO cells had similar percentages of cilia with Smo after 12-h and 24-h SAG stimulation, indicating that lack of satellites directly or indirectly caused a delay in translocation of Smo to the cilium in response to SAG (Fig 6A). Reduction in Smo relocalization phenotype was rescued by stable expression of LAP-tagged full-length human PCM1 at 4 and 8 h time points of SAG stimulation (Fig 6A). In the IMCD3 PCM1 KO and control cells that formed cilia, we determined ciliary Smo levels and concentrations, and found reduced localization in IMCD3 PCM1 KO cells relative to control cells. Ciliary SMO level in IMCD3 PCM1 KO cells decreased to $0.22 \pm 0.16$ compared to control cells, which is normalized to 1 after 4-h SAG treatment (Fig 6B). Ciliary SMO concentration decreased to $0.19 \pm 0.18$ in IMCD3 PCM1 KO cells compared to control cells normalized to 1. This reduction was not due to changes in the cellular abundance of Smo (Fig 6C). To assay the transcriptional response to Hedgehog pathway activation,

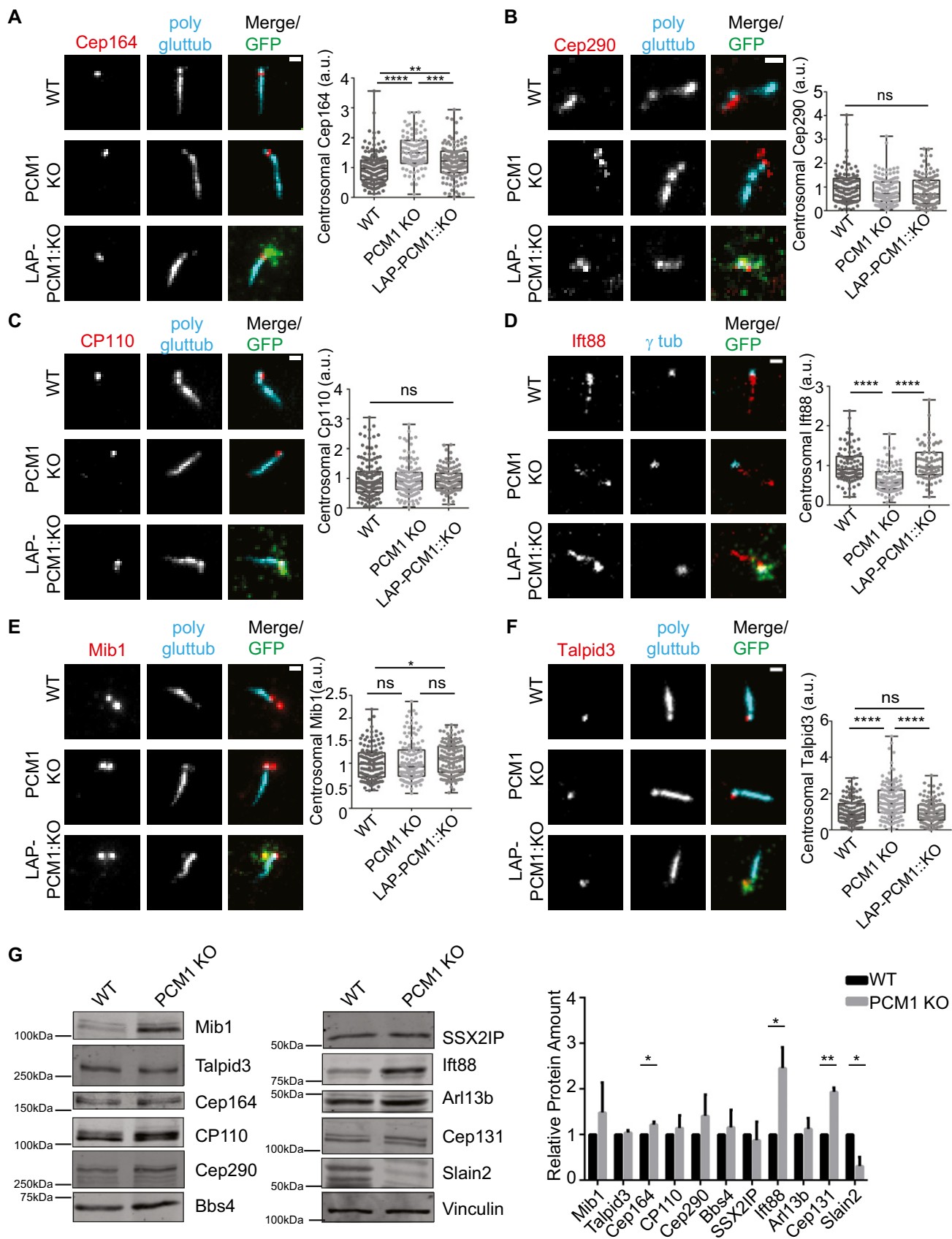

Figure 4.

◄

**Figure 4.   Satellites have varying degrees of effects on regulating centrosomal and cellular abundance of key ciliogenesis factors.**

A–F   Effect of satellite loss on centrosomal abundance of proteins in cells serum-starved for 24 h. Control, IMCD3 KO, and IMCD3 KO stably expressing LAP-PCM1 cells were fixed and stained with antibodies for the indicated proteins along with anti-GFP for marking LAP-PCM1 and anti-polyglutamylated tubulin or anti-gamma-tubulin for marking centrosomes. DNA was stained with DAPI. Scale bar, 1 μm. Both ciliated and unciliated cells were quantified in a blinded manner. Images represent cells from the same coverslip taken with the same camera settings. The centrosomal fluorescence intensity of the indicated proteins was measured in a 3 μm$^2$ square area around the centrosome, and levels are normalized to 1. Results shown are the mean of two independent experiments ± SD (100 cells/experiment, *$P < 0.5$, **$P < 0.01$, ***$P < 0.001$, ns: not significant *t*-test). Error bars represent SD. Horizontal line represents mean value of each group. The boxes include data from the 25$^{th}$ to 75$^{th}$ percentiles in each group.

G      Effects of satellite loss on cellular abundance of centrosome proteins. Whole-cell lysates from control and IMCD3 PCM1 KO cells serum-starved for 24 h were immunoblotted with the indicated antibodies. Vinculin was used as a loading control. Results shown are the mean of two independent experiments ± SD (*$P < 0.5$, **$P < 0.01$, *t*-test).

we examined the expression of the Hedgehog target gene *Gli1* in control and PCM1 KO cells. While wild-type cells had robust activation of Gli1 expression (normalized to 100%), PCM1 KO cells failed to upregulate Gli1 expression at 24 h (35% ± 25.6; Fig 6D). There was a very small but significant decrease in Gli1 expression in PCM1 KO cells (89.46% ± 5.41) relative to control cells (100%) before SAG stimulation (Fig 6E). Taken together, these results indicate that satellites are required for the localization of sufficient levels of Smo at cilia, and efficient activation of the Hedgehog pathway.

## Satellites are required for epithelial cell organization in 3D cultures

When grown in a three-dimensional gel matrix, epithelial cells organize into polarized, spheroid structures that reflect the *in vivo* organization of the epithelial tissues. Epithelial spheroids have been widely used to assay cilia dysfunction, because proper cilium assembly and ciliary signaling is essential for the establishment of the highly organized architecture and apicobasal polarity of epithelial cells in 3D [53–55]. To assay the consequences of ciliary defects associated with loss of satellites on tissue architecture, we used the 3D spheroid cultures of IMCD3 cells that mimic *in vivo* organization of the kidney collecting duct [56]. Control and IMCD3 PCM1 KO cells were grown in Matrigel for 3 days, serum-starved for 2 days, and spheroid architecture was visualized by staining cells for markers for cilia, cell–cell contacts, and polarity. Control cells formed spheroids with prominent lumen, apical cilia oriented toward the lumen, and proper organization of apical and basal surfaces (ZO-1 and beta-catenin; Fig 7A). However, PCM1 KO cells had a significant decrease in their ability to form proper spheroids relative to control cells, which was rescued by transient expression of myc-PCM1 (control: %70.75 ± 3.89; PCM1 KO: %46.45 ± 1.91; rescue: %59.80 ± 1.31; Fig 7A and B). Instead, they formed cell clusters with no lumens that had misoriented cilia and disorganized apical and basolateral junctions (Fig 7A). Analogous to the ciliogenesis defects of 2D serum-starved cultures, ciliogenesis efficiency was significantly reduced in IMCD3 PCM1 KO cells grown in 3D cultures and this defect was rescued by transient expression of myc-PCM1 (control: %64.08 ± 19.19; PCM1 KO: %36.98 ± 21.28; rescue: %61.81 ± 17.25; Fig 7A and C). Expression of myc PCM1 in rescue experiments was confirmed by immunofluorescence experiments (Appendix Fig S1). In agreement with the function of satellites in ciliogenesis and signaling, these results identify important roles for satellites in epithelial organization in this *in vitro* tissue model.

## Loss of satellites leads to a rearrangement of the global proteome profile, but not the global transcriptome profile

Loss of satellites variably affected the cellular abundance of centrosome and cilium proteins in IMCD3 (Figs 4 and EV3) and in RPE1 cells as reported previously [36]. These data suggest functions for satellites in regulating the cellular abundance of proteins. However, whether these changes are a consequence of transcriptomic or proteomic modulation and which centrosome proteins are affected by these changes have not been investigated. To address this, we performed a systematic comparative analysis of the global transcriptomic and proteomic profile of IMCD3 PCM1 KO cells.

RNA sequencing (RNAseq) analysis of PCM1 KO and control IMCD3 cells revealed 8 genes, which were differentially regulated with log$_2$ fold change > 1.5 and $P < 0.05$ (Dataset EV1). Gene ontology and KEGG pathway functional analysis did not identify any functional clustering among these genes. Therefore, it seems unlikely that the differential expression of the identified genes is a direct result of the loss of satellites and the direct cause of the phenotypes observed in PCM1 KO cells. Together, these results show that loss of satellites does not induce a major stress response or compensation mechanism in these cells.

In contrast to the transcriptome analysis, tandem mass tag (TMT) labeling-based quantitative analysis of the global proteome of control and IMCD3 PCM1 KO cells showed that loss of satellites substantially altered the proteomics profile. A total of 6,956 proteins were identified with high confidence, 295 of which were upregulated, while 296 were downregulated (log$_2$-transformed normalized ratios > 0.5) (Fig 8A, Dataset EV2 and EV3). GO-term analysis of the major biological processes and compartments enriched or depleted significantly in PCM1 KO cells identified categories related to "centrosome", "cell proliferation", and "microtubules" (Fisher's exact test, FDR < 0.05; Fig EV5). This is in agreement with the function of satellites as regulators of the centrosome/cilium complex. Moreover, the GO terms actin cytoskeleton, cell migration and adhesion, endocytosis, neuronal processes, metabolic processes were among the most enriched and depleted compartments and processes in Fig EV5C and D, which was also illustrated as functional network clustering analysis in Fig EV5A and B. Together, this analysis suggests possible functions for satellites in these processes.

To specifically determine the consequences of satellite loss on the cellular abundance of centrosome proteins, we cross-correlated the global proteome dataset with the published centrosome proteome [57]. Unexpectedly, we found that majority of the 116 centrosome proteome identified remains in place, where only a few proteins including the satellite protein Cep131 and regulatory

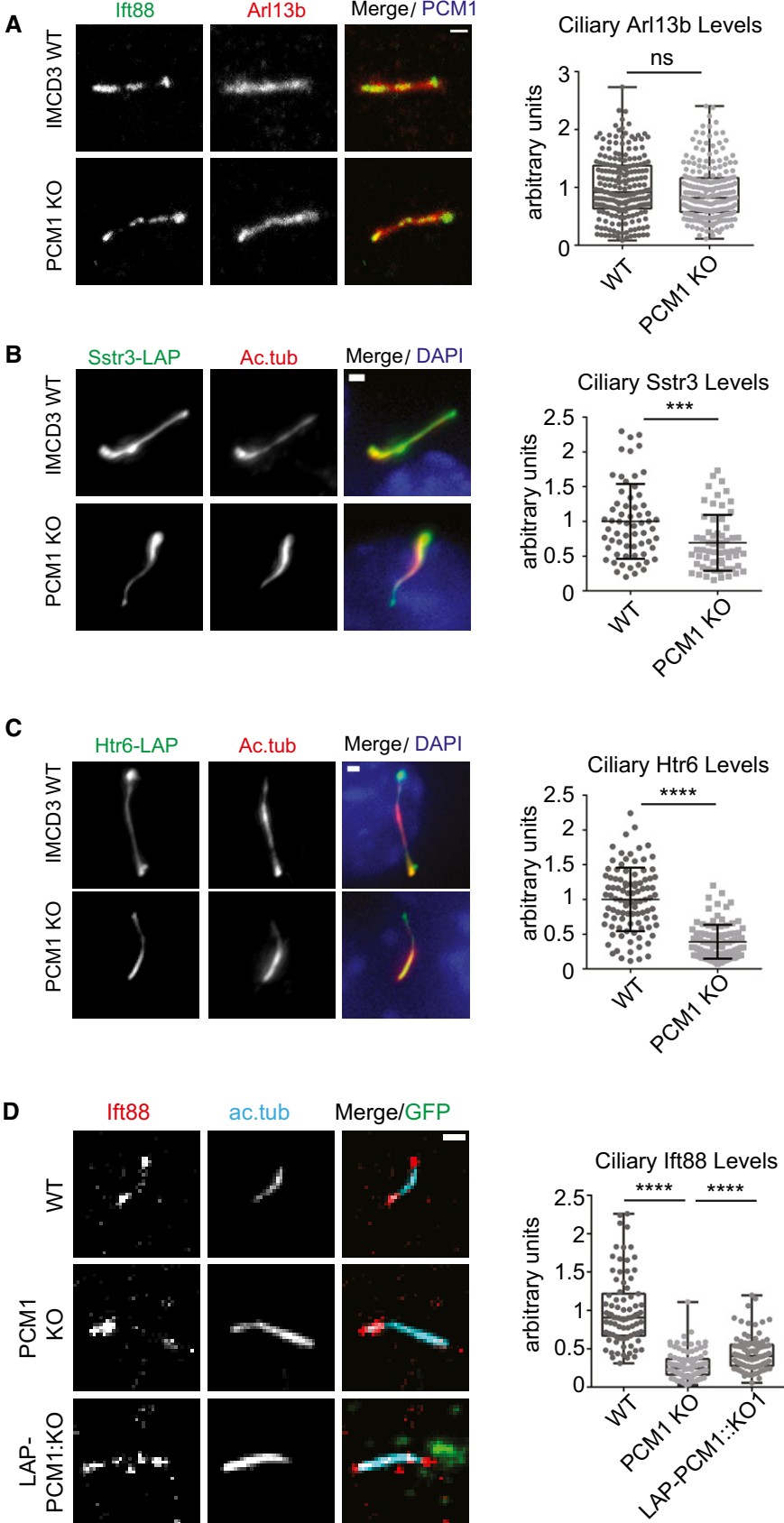

**Figure 5.**

◄

**Figure 5.  Satellites are required for regulating the ciliary content.**

A–C   Effects of satellite loss on the ciliary recruitment of ciliary membrane and shaft proteins. Control and IMCD3 PCM1 KO cells were serum-starved for 24 h, fixed and stained for the cilium with anti-acetylated tubulin (Ac. tub) or Arl13b, and (A) Ift88 with anti-Ift88 and Arl13b with anti-Arl13b (B) SSTR3-LAP with anti-GFP, (C) HTR6-LAP with anti-GFP.

D    Effect of satellite loss on the ciliary abundance of Ift88 in control, IMCD3 PCM1 KO, and IMCD3 PCM1 KO::LAP-PCM1 cells serum-starved for 24 h. Cells were fixed and stained with Ift88, the ciliary marker acetylated tubulin. Scale bar, 1 μm. Images represent cells from the same coverslip taken with the same camera settings. Relative quantifications of the ciliary intensity of the indicated proteins are shown as the mean of two independent experiments ± SD (100 cells/experiment, **$P < 0.01$, ***$P < 0.001$, ****$P < 0.0001$, ns: not significant, $t$-test). The mean intensity of the control proteins was normalized to 1. Horizontal line represents the mean value of each group. The boxes include data from the 25th to 75th percentiles of each group.

proteins Ran1Gap2 and Prkar2B were perturbed in the global proteome (Fig 8B–D) [34,58–60]. Consistent with the proteomics data, a significant increase in the cellular abundance of Cep131 and a significant decrease in SLAIN2 was detected using immunoblotting experiments (Fig 4G). Due to the low abundant nature of centrosome proteins, we set a lower significance threshold ($\log_2 > 0.3$ for TMT labeling) for detecting changes in abundance, which identified 21 proteins in IMCD3 KO cells to be differentially expressed. The upregulated and downregulated centrosome proteins were implicated in a wide range of processes including suppressors and activators of ciliogenesis, centriole duplication, spindle formation, signaling, and microtubule cytoskeleton (Fig EV5C and D). The lack of functional clustering among these proteins shows that the regulatory roles of satellites on their resident proteins are specific to each protein and that phenotypic consequences of loss of satellites are likely combinatorial effects of these changes, instead of a specific pathway.

# Discussion

The results of our study reveal two specific roles of satellites in vertebrate cells. First, satellites are required for efficient ciliogenesis, regulation of the ciliary shaft and the ciliary membrane content, and epithelial cell organization in 3D cultures. In particular, satellites are required directly or indirectly for efficient response to Hedgehog pathway agonists. Second, satellites are regulators of the protein targeting and cellular proteostasis, not only for centrosome proteins but also for proteins from a diverse set of pathways. Given that mutations affecting various satellite components cause ciliopathies [5,61] and loss of satellites in zebrafish leads to ciliopathy-related phenotypes [20], our findings suggest variable degrees of defects in cilium assembly, signaling, and tissue architecture linked to satellite mutations as mechanisms underlying diverse disease phenotypes of ciliopathies.

A variety of satellite-specific functions were reported through previous loss-of-function studies targeting PCM1. Transient depletion of PCM1 caused defects in organization of the interphase microtubule network in U2OS cells [37], cell cycle progression and cell proliferation in MRC-5 primary human fibroblasts and HeLa cells [62], and cilium formation in RPE1 cells [39,40]. Constitutive depletion of PCM1 from RPE1 and GBM cells caused inhibition of ciliogenesis [36,42]. We found that satellites are only required for cilium assembly and function, but not for cell proliferation, cell cycle progression, and centriole duplication in IMCD3 and RPE1 cells. This variation in phenotypes, except for the cilium-related ones, could be due to a functional compensation mechanism that rescues the defects caused by constitutive loss of satellites, previously

reported for another satellite protein Azi1 [63]. Alternatively, the requirement for satellites in these processes may simply vary in different cell types, as we show for cilium-related processes in this study. We would like to highlight that loss-of-function studies of PCM1 also defined functions for satellites in autophagy [30], stress response [21,23], and neuronal progenitor cell proliferation and migration [38,41,64], which we have not focused on in this study.

Satellite loss in RPE1 cells caused an almost complete inhibition of ciliogenesis, identifying a direct and requisite role for satellites in cilium assembly in these cells [36]. In contrast, satellite-less IMCD3 cells had a much weaker twofold reduction in their ability to ciliate. The variation in the phenotypes observed in IMCD3 and RPE1 cells might be due to differences in the species and tissues they are derived from, ciliogenesis mechanisms, and mechanism of transformation. IMCD3 cells that ciliated in the absence of satellites had cilia of similar length compared to control cells, which suggests that satellites in these cells are required to initiate assembly of the ciliary axoneme. While satellite-less cells proceeded normally through early steps of mother centriole maturation, they were defective in the recruitment of Ift88 to the basal body, suggesting that satellites promote ciliogenesis upstream of this stage during ciliogenesis. Supporting the relationship between IFT machinery and satellites, proteomics studies identified putative interactions of satellites with various IFT-A and IFT-B components, supporting the functional relationship between the IFT machinery and satellites [27,65,66]. Of note, we observed that centrosomal levels of Cep164 and Talpid3 increased in satellite-less cells and we do not know whether these changes have any functional consequences on ciliogenesis.

Satellites were shown to regulate the stability of the ciliogenesis factor Talpid3 and the Atg8 autophagy marker Gabarap through sequestration of Mib1, which suggested that they might mediate functions as regulators of protein degradation pathways [30,36]. However, the turnover of which centrosome proteins are regulated by satellites was not known. Global proteomics of IMCD3 cells showed that the majority of the centrosome proteome remained unaltered and that only the expression of a small subset of centrosome proteins were perturbed. Importantly, Mib1 was among upregulated centrosome proteins in IMCD3 satellite-less cells, corroborating the negative crosstalk between PCM1 and Mib1 stability [23,30,36,67]. It is important to note that the global proteome analysis is limited in identifying all the centrosome proteins and determining changes in their expression levels with high sensitivity, due to the low abundance levels of most centrosome proteins in cells [68].

In addition to regulating protein stability, satellites were shown to mediate their functions through targeting proteins to the centrosome or sequestering them to limit their centrosomal recruitment.

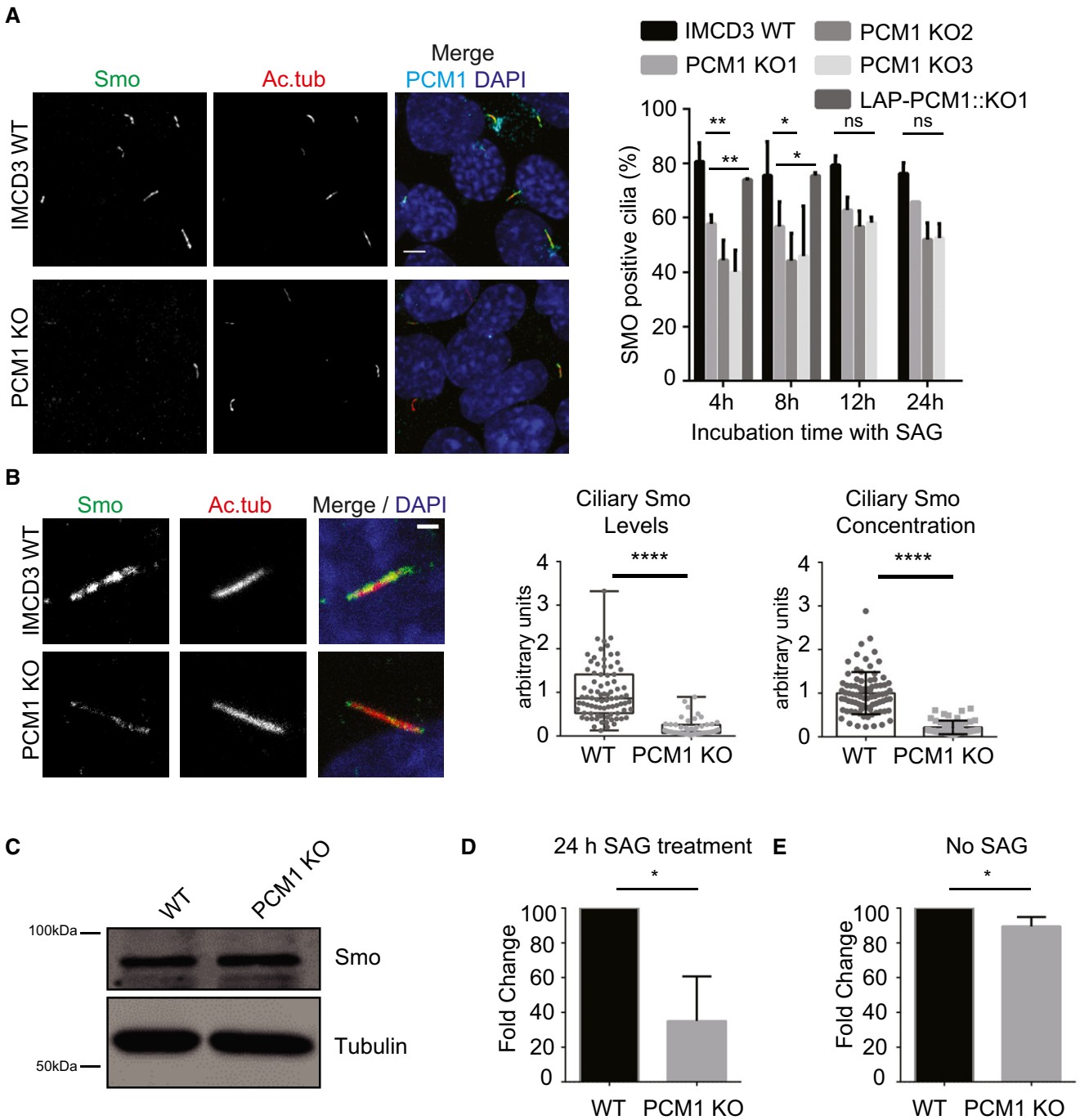

**Figure 6. Satellites are required for ciliary Smo recruitment and Gli1 transcriptional activation in response to Hedgehog signals.**

A    Effect of satellite loss on ciliary recruitment of Smo. Control, IMCD3 KO, and IMCD3 KO stably expressing LAP-PCM1 cells were serum-starved for 24 h, treated with 200 nM SAG for the indicated times, fixed and stained for Smo, acetylated tubulin (Ac. tub), and DAPI. Percentage of Smo-positive cilia was quantified. Scale bar, 4 μm. Results shown are the mean of three independent experiments ± SD (250 cells/experiment, **P < 0.01, *P < 0.05, t-test).

B    Effect of satellite loss on ciliary SMO levels. The ciliary intensity of SMO was quantified by staining 4 h SAG-treated cells for Smo and acetylated tubulin. DNA was stained with DAPI. Scale bar is 4 μm. Relative quantifications of the ciliary intensity and ciliary concentration of SMO are shown as the mean of two independent experiments ± SD (100 cells/experiment, ****P < 0.0001 t-test).

C    Effect of satellite loss on cellular abundance of Smo. Whole-cell lysates from control and IMCD3 PCM1 KO cells were immunoblotted with Smo and alpha-tubulin (loading control). Smo cellular levels were similar between control and IMCD3 PCM1 KO cells.

D, E    Effect of satellite loss on Gli1 transcriptional activation. (D) Gli1 mRNA was quantified by qPCR before SAG treatment and 24 h after SAG treatment, and its fold change is normalized to control cells (=100). Results shown are the mean of two independent experiments ± SD (*P < 0.05, t-test). (E) Cellular abundance of Gli1 mRNA in control and PCM1 KO cells. Effect of satellite loss on Gli1 transcriptional activation. Gli2 mRNA was quantified by qPCR without any SAG treatment. Relative mRNA amount is normalized to control cells (=100). Results shown are the mean of two independent experiments ± SD (*P < 0.05, t-test).

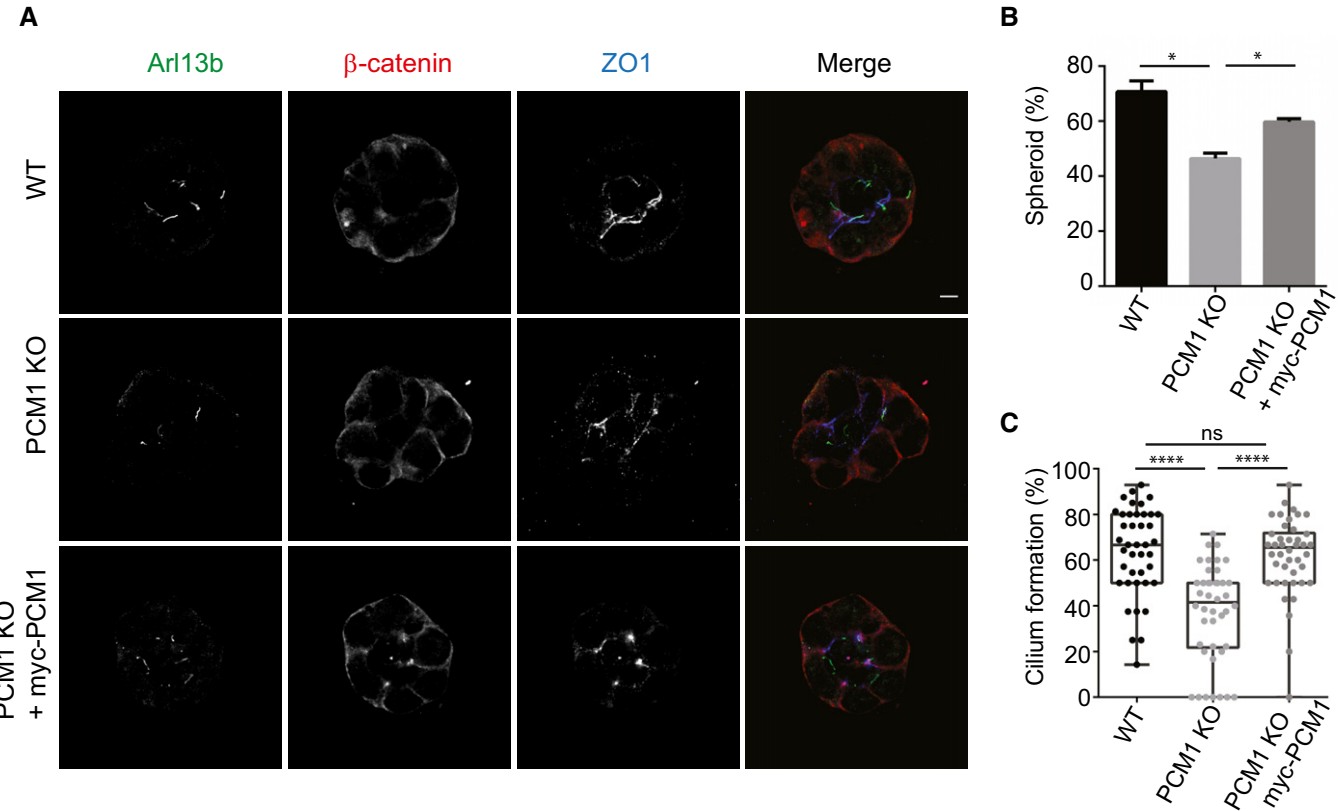

**Figure 7. Satellites are required for epithelial cell organization in 3D spheroid cultures.**

A  Effects of satellite loss on epithelial cell organization. IMCD3 cells were grown in Matrigel for 3 days and serum-starved for 2 days. The spheroids were fixed and stained for apical junctions with anti-ZO1, cilia with anti-Arl13b, and basolateral surfaces with anti-beta-catenin. Scale bar, 5 μm.

B  Quantification of the frequency of spheroid formation in control, IMCD3 PCM1 KO cells, and IMCD3 PCM1 KO cells transfected with myc-PCM1. Results shown are the mean of three independent experiments ± SD (50 spheroids/experiment, *P < 0.05, t-test).

C  Quantification of the cilia formation in spheroids in control, IMCD3 PCM1 KO cells, and IMCD3 PCM1 KO cells transfected with myc-PCM1. Results shown are the mean of three independent experiments ± SD (50 spheroids/experiment, ****P < 0.00001, t-test). Error bars represent SD. Horizontal line represents the mean value of each group. The boxes include data from the 25th to 75th percentiles in each group.

In agreement, we detected changes in the centrosomal and ciliary levels of key ciliogenesis proteins, as well as ciliary membrane and shaft proteins. These results show functions for PCM1 in the ciliary recruitment of proteins for the first time and also suggest that these defects might be the underlying the satellite-related defects *in vivo* and in disease. Notably, for some proteins like Ift88 and Mib1, we showed that changes in the centrosomal levels of these proteins do not scale with changes in their cellular abundance. An attractive possibility for this discrepancy is that satellite-less cells compensate for functional defects through modulating the centrosomal recruitment of proteins. Because we performed this analysis only for a handful of proteins, future studies like identification of the centrosome and cilium proteome in satellite-less cells are required to elucidate the complete list of centrosome and cilium proteins regulated by PCM1.

Despite the unaltered abundance of the centrosome proteome, loss of satellites led to a significant rearrangement of the global proteome, suggesting that satellites regulate cellular proteostasis. Importantly, these global studies provided insight into previously uncharacterized satellite functions. Among the most enriched

pathways were the ones related to the actin cytoskeleton such as cell migration, cell adhesion, and endocytosis. The possible functional connection of satellites to actin-related processes is noteworthy due to the function of centrosomes as actin-nucleating centers. In particular, satellites are required for centrosomal actin nucleation through targeting actin assembly factors to the centrosome [69,70]. Satellites might also function in other actin-regulated ciliary processes, such as periciliary vesicle transport to the basal body and cilia length regulation [71–74]. Moreover, we also found a markedly strong increase at the protein level in neurogenesis-linked pathways, such as neuronal body and extensions, postsynaptic membrane, and axon guidance. Consistently, mutations in PCM1 were associated with schizophrenia [75–77], and PCM1, Hook3, DISC1, and SDCCAG8 were implicated in maintaining neural progenitor cells and neuronal migration during cortical development. Future studies on studying satellites in different contexts will be important to identify the full range of satellite functions.

While phenotypic characterization of stable genetic knockout model for of satellite-less cells has led to important insights into satellite function and molecular mechanism, its disadvantage is the

**A**

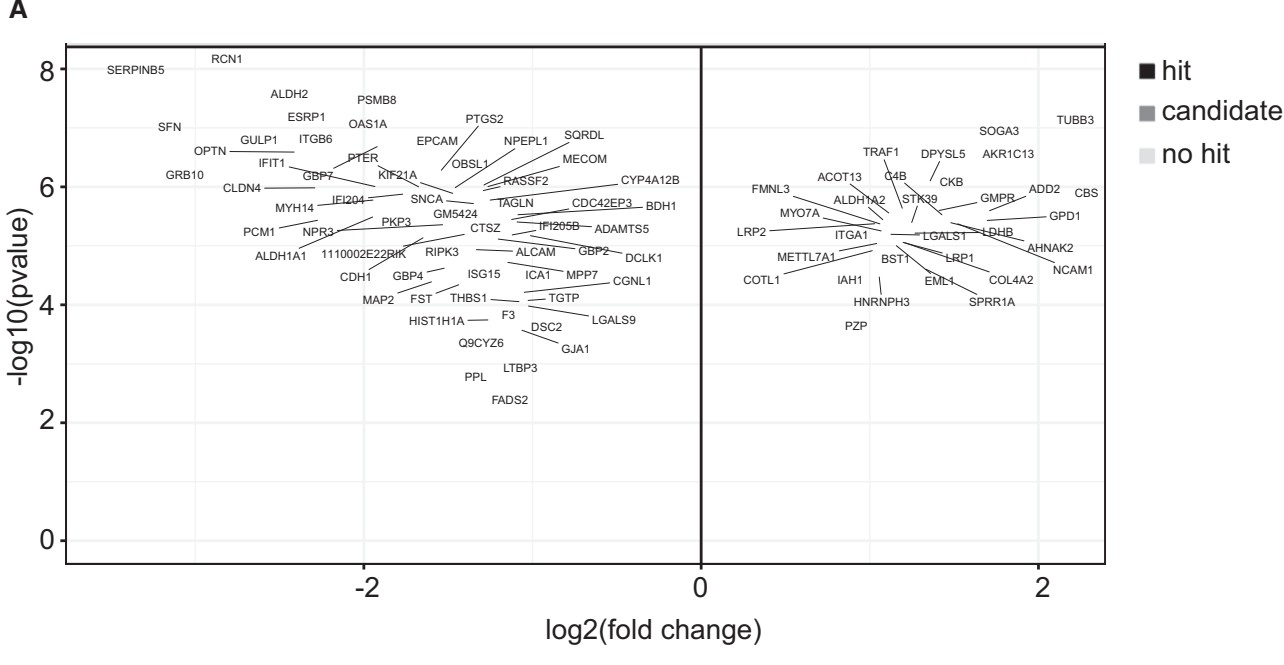

**B**

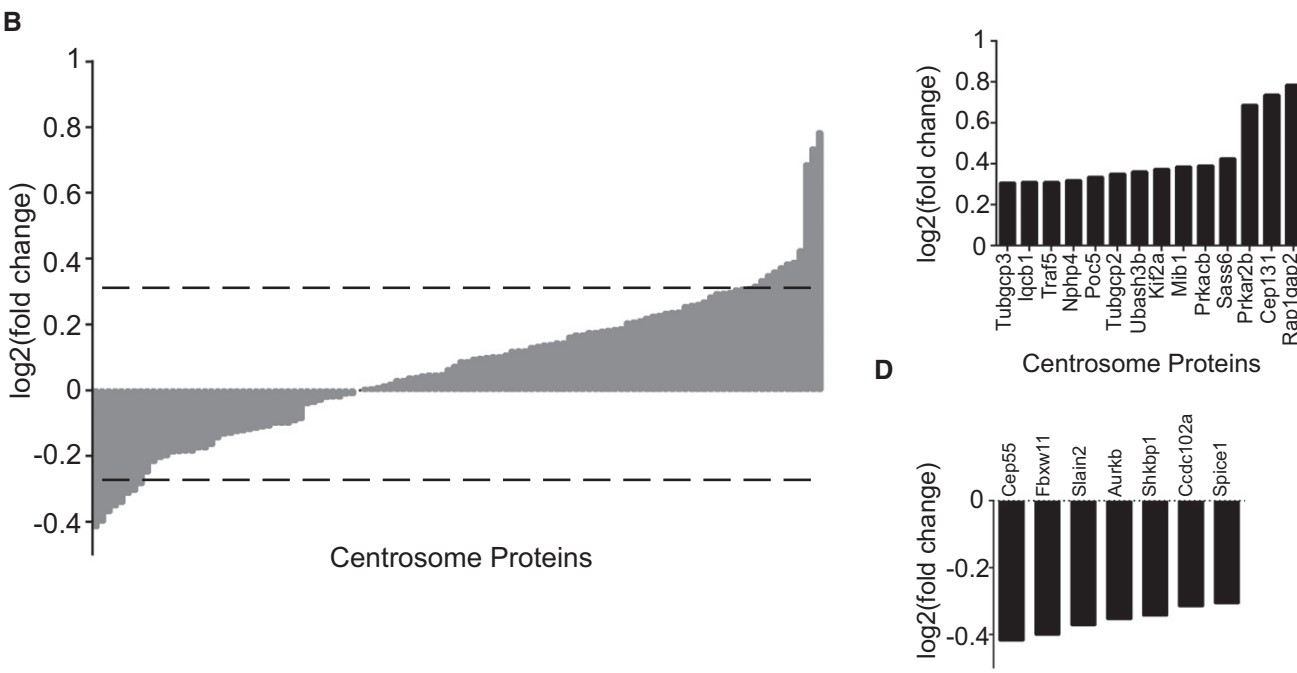

**Figure 8. Loss of satellites changes the global proteome.**

Quantitative proteomics profiling of IMCD3 PCM1 KO cells.

A Volcano plots illustrating that 591 proteins show a significant change in abundance. Data were derived from mass spectrometry analysis of three biological replicates of control and IMCD3 PCM1 KO cells. The x-axis corresponds to the $\log_2$ fold change value, and the y-axis displays the $-\log_{10}$ P-value. Proteins were classified into hits (indicated with black circles) if the FDR is smaller than 0.05 and a fold change of at least twofold is observed. Proteins were classified into candidates (indicated with medium gray circles) if the FDR is smaller than 0.2 and a fold change of at least 1.5-fold is observed.

B Comparison of the centrosome proteome with the IMCD3 PCM1 KO global proteome. 112 centrosome proteins identified in the global proteome (x-axis) were plotted against their $\log_2$-transformed ratios of fold changes (y-axis) between IMCD3 PCM1 KO and control samples. Dashed line indicates the threshold we determined for the significantly upregulated and downregulated proteins represented in (C and D).

C Significantly upregulated centrosome proteins ($\log_2 > 0.3$).

D Significantly downregulated centrosome proteins ($\log_2 < -0.3$).

possible masking of some satellite phenotypes. Generation and maintenance of PCM1 KO lines using the CRISPR/Cas9 genome editing approach requires a long time frame where cells undergo many successive divisions without satellites and thus might have enough time to activate compensatory mechanisms that can mask satellite-specific phenotypes, specifically through proteomic modulation for satellite-less cells as their transcriptome scarcely change [78,79]. Likewise, some of the changes we detected in global transcriptomics and proteomics might be specific to cell clones. Future studies using spatially and temporally controlled *in vitro* and *in vivo* experiments are required to overcome these limitations.

In summary, our findings identify a more complex molecular and functional relationship between satellites and the centrosome/cilium complex. We suggest that satellites are among the additional regulatory mechanisms vertebrates acquired during evolution in order to prevent defects in the biogenesis and function of the centrosome/cilium complex and thereby associated diseases. Future studies are required to elucidate the functions of satellites in different cell types and tissues and to investigate how defects in these functions are associated with different disease states.

# Materials and Methods

## Cell culture and transfection

IMCD3::Flippin cells (gift from Max Nachury, UCSF, CA) and hTERT::RPE1 cells (ATCC® CRL-4000) were grown in DMEM/F12 50/50 (Life Technologies) supplemented with 10% fetal bovine serum (FBS; Life Technologies). HEK293T cells were grown in DMEM supplemented with 10% FBS. Media were supplemented with 100 U/ml penicillin and 100 μg/ml streptomycin (Life Technologies). All cells were cultured at 37°C and 5% $CO_2$. All cell lines were authenticated by Multiplex Cell line Authentication (MCA) and were tested for mycoplasma contamination by MycoAlert Mycoplasma Detection Kit (Lonza). RPE1 and IMCD3 cells were transfected with the plasmids using Lipofectamine LTX according to the manufacturer's instructions (Life Technologies). HEK293T cells were transfected with the plasmids using 1 μg/μl polyethylenimine, MW 25 kDa (PEI, Sigma-Aldrich, St. Louis, MO). IMCD3 cells stably expressing HTR6-LAP, SSTR3-LAP, LAP-PCM1, and LAP-PCM1 (1–3,600) were generated by cotransfecting IMCD3 Flip-in cells with pEF5.FRT.LAP expression vectors and pOG44 at a ratio of 1:7 using Lipofectamine LTX. After incubating cells with the transfection mix for 6 h, medium was replaced with complete medium for 24 h, which was followed by selection with 200 μg/ml hygromycin B (Sigma-Aldrich) for 1 week. Colonies formed within 10–14 days, individual colonies were picked after selection and screened by immunofluorescence for expression of LAP-fusion proteins. For cilium-related experiments, cells were incubated in low-serum medium (0.5% serum) for 24 h or 48 h. For Smo relocalization experiments, cells were serum-starved for 24 h and incubated with 200 nm SAG (EMD Millipore) for 4, 8, 12, or 24 h.

## 3D spheroid assay

40 μl/well of 100% Matrigel (BD Biosciences) was solidified in Lab-Tek 8-well chamber slides (Thermo Fisher) through 15-min incubation at 37°C. Control and IMCD3 PCM1 KO cells were trypsinized, washed with PBS, and resuspended in DMEM/F12 50/50 supplemented with 2% fetal bovine serum and 2% Matrigel, and 5,000 cells/well were plated in Matrigel-coated 8-well chamber slides. Cells were grown at 37°C for 4 days, serum-starved for 1 day, fixed with paraformaldehyde (PFA) and stained with the indicated markers. Cell clusters with no hollow lumen, multiple lumens, disturbed organization of apical and basal markers, and/or misaligned nuclei were quantified as defective. Rescue experiments were performed by transient transfection of myc-BirA*-PCM1.

## Lentivirus production and cell transduction

Lentivirus were generated as described [80], using pLentiCRISPRv2, pLentiCRISPRv2-mousePCM1, and pLentiCRISPRv2-humanPCM1 plasmids as transfer vectors. For infection, $1 \times 10^5$ IMCD3 cells were seeded on 6-well tissue culture plates the day before infection, which were infected with 1 ml of viral supernatant the following day. 24 h post-infection, medium was replaced with complete medium. 48 h post-infection, cells were split and selected in the presence of 2 μg/ml puromycin for 5–7 days till all the control cells died. LentiCRISPRv2 empty vector-transfected and puromycin-selected cells were used as control. After the PCM1 knockout efficiency was determined in the heterogeneous pool using immunofluorescence experiments, cells were trypsinized and serial dilutions were performed into normal growth medium. Colonies formed within 10–14 days were then trypsinized and expanded for screening for knockouts by immunofluorescence and Western blotting. IMCD3 cells stably expressing mCherry-H2B were generated by infection of cells with mCherry-H2B-expressing lentivirus. Lentivirus was generated using pLVPT2-mCherry-H2B plasmid as transfer vectors.

## Cloning and genome editing

Full-length cDNA of BBS4 (NM_001252678) was obtained from DF/HCC DNA Resource Core (Harvard Medical School, MA). Full-length cDNAs for human PCM1 (NM_006197), HTR6 (NM_021358), and SSTR3 (BC120843) were amplified from the peGFP-N1-PCM1, peGFP-N1-SSTR3, and peGFP-N1-HTR6 plasmids. PCR-amplified open reading frames of full-length PCM1 and its deletion mutant (1–3,600), HTR6, BBS4, and SSTR3 were cloned into pDONR221 using the Invitrogen Gateway system. Subsequent Gateway recombination reactions using pMN444 pEF5-FRT-LAP DEST, provided by M. Nachury (UCSF, CA), were used to generate LAP-PCM1, LAP-PCM1 (1–1,200), HTR6-LAP, SSTR3-LAP, and BBS4-LAP. Full-length PCM1 was PCR-amplified using forward primer flanked with XhoI Site and reverse primer flanked with KpnI Site and ligated into pcDNA3.1-mycBirA* plasmid digested with XhoI and KpnI using T4 DNA ligase. Guide RNAs targeting human and mouse PCM1 coding exon 2 were cloned into LentiCRISPRv2 (gift of Feng Zhang, Addgene plasmid #52961) using the following primers: mouseOligo forward: 5′-CACCGCTGCTGTGTGGAAACGTATG-3′, mouseOligo reverse: 5′- AAACCATACGTTTCCACACAGCAGC-3′, humanOligo forward: 5′- CACCGCTACTGTGTGGGAACGTATG-3′, humanOligo reverse: 5′- AAACCATACGTTCCCACACAGTAGC-3′, nontargeting forward: 5′- CGCTTCCGCGGCCCGTTCAA-3′, nontargeting reverse: 5′-TTGAACGGGCCGCGGAAGCG-3′. All plasmids were verified by DNA sequencing.

## Flow cytometry and proliferation assays

Cells were harvested and fixed in 70% ethanol at −20°C overnight, followed by washes in phosphate-buffered saline (PBS) and staining with 40 μg/ml propidium iodide and 10 μg g/ml RNaseA at 37 for 30 min. Cell cycle analysis was performed using a Sony SH800S Cell Sorter for IMCD3 cells (Sony) and BD Accuri C6 Sorter for RPE1 cells (BD Biosciences). For proliferation assays, $2 \times 10^5$ cells were plated and cells were counted at 1, 2, 3, and 5 days. At each count, cells were split at 1:2.

## Cell lysis and immunoblotting

For preparation of cell extracts, cells were trypsinized off the plate, washed in PBS, and lysed in lysis buffer (50 mM Tris (pH 7.6), 150 mM NaCI, 1% Triton X-100, and protease inhibitors) for 30 min at 4°C and centrifuged at 15,000 g for 15 min. The protein concentration of the resulting supernatants was determined with the Bradford solution (Bio-Rad Laboratories, CA, USA). For immunoblotting, equivalent amount of total protein for each cell extract were first resolved on SDS–PAGE gels, transferred onto nitrocellulose membranes, blocked in TBS-T (Tris-buffered saline with 0.1% Tween-20) with 5% milk, blotted with primary antibodies overnight at 4°C and secondary antibodies for 1 h at room temperature. Visualization of the blots was carried out with the LI-COR Odyssey® Infrared Imaging System and software at 169 μm (LI-COR Biosciences).

## RNA isolation, cDNA Synthesis, and qPCR

Total RNA was isolated from control and IMCD PCM1 KO cells before SAG treatment and 24 h after SAG treatment using NucleoSpin RNA kit (Macherey-Nagel) according to the manufacturer's protocol. Quantity and purity of RNA were determined by measuring the optical density at 260 and 280 nm. Single-strand cDNA synthesis was carried out with 1 mg of total RNA using iProof High-Fidelity DNA Polymerase) qPCR analysis of Gli1 was performed with primers 5′ GCATGGGAACAGAAGGACTTTC 3′ and 5′ CCTGGGACCCTGACATAAAGTT 3′ using GoTaq® qPCR Master Mix (Promega).

## Antibodies

Anti-PCM1 antibody was generated and used for immunofluorescence as previously described [29]. Other primary antibodies used for immunofluorescence in this study were rabbit anti-PCM1 (Proteintech—19856-1-AP) at 1:1,000, goat anti-PCM1 (E-5, Santa Cruz Biotechnology—sc50164) at 1:1,000, mouse anti-γ-tubulin (GTU-88; Sigma-Aldrich—T6557) at 1:4,000, mouse anti-GFP (3e6; Invitrogen—11120) at 1:750, mouse anti-acetylated tubulin (6-11-B; Sigma-Aldrich—T6793)) at 1:10,000, mouse anti-polyglutamylated tubulin (GT335, Adipogen—AG-20B-0020) at 1:500, mouse anti-Smo (Santa Cruz Biotechnology—sc166685) at 1:500, mouse anti-Arl13b (Neuromab—75-287) at 1:1,000, rabbit anti-IFT88 (Proteintech—13967-1-AP) at 1:200 [81], mouse anti-MIB1 (Santa Cruz Biotechnology—sc-393811) at 1:500, rabbit anti-CP110 (Proteintech—12780-1-AP) at 1:200 [82], rabbit anti-Talpid3 (Proteintech—24421-1-AP) at 1:500 [83], rabbit anti-Cep290 (Abcam—ab84870) at

1:750 [84], rat anti-ZO1 (Santa Cruz Biotechnology—sc33725) at 1:200, rabbit anti-beta-catenin (Proteintech—51062-2-AP) at 1:500, and mouse anti-Cep164 at 1:500 [85]. Secondary antibodies used for immunofluorescence experiments were Alexa Fluor 488-, 568-, or 633-coupled (Life Technologies), and they were used at 1:2,000. Primary antibodies used for Western blotting were rabbit anti-PCM1 (ProteinTech—19856-1-AP) at 1:500, mouse anti-γ-tubulin (GTU-88; Sigma-Aldrich—T6557) at 1:4,000, mouse anti-acetylated tubulin (6-11-B; Sigma-Aldrich) at 1:10,000, mouse anti-polyglutamylated tubulin (GT335, Adipogen) at 1:500, mouse anti-Arl13b (Neuromab—75-287) at 1:500, mouse anti-Smo (D-19, Santa Cruz Biotechnology—sc166685) at 1:500, rabbit anti-IFT88 (Proteintech—13967-1-AP) at 1:500, mouse anti-MIB1 (Sigma-Aldrich—M5948) at 1:500 [36], rabbit anti-CP110 (Proteintech—12780-1-AP) at 1:500 [82] and rabbit anti-Cep164 (Proteintech—22227-1-AP) at 1:500, rabbit anti-Talpid3 (Proteintech—24421-1-AP) at 1:500, rabbit anti-Cep290 (Proteintech—22490-1-AP) at 1:500 rabbit, rabbit anti-BBS4 (Proteintech—12766-1-AP) at 1:500 [86], and mouse anti-vinculin (H-10—Santa Cruz Biotechnology—sc-25336) at 1:1,000. Secondary antibodies used for Western blotting experiments were IRDye 680- and IRDye 800-coupled and were used at 1:15,000 (LI-COR Biosciences).

## Immunofluorescence, microscopy, and quantitation

For immunofluorescence experiments, cells were grown on coverslips and fixed in either methanol for 5 min or 4% PFA for 15 min in PBS for indirect immunofluorescence. For CP110 staining, cells were pre-extracted in 0.01% Tx-100 in PBS for 1 min before fixation. After rehydration in PBS, cells were blocked in 3% BSA (Sigma-Aldrich) in PBS + 0.1% Triton X-100 for 30 min. Coverslips were incubated in primary antibodies diluted in blocking solution for 1 h at room temperature and Alexa Fluor 488-, 594-, or 680-conjugated secondary antibodies for 1 h at room temperature. Samples were mounted using Mowiol mounting medium containing N-propyl gallate (Sigma-Aldrich). Coverslips of cells were imaged using LAS X software (Premium; Leica) on a scanning confocal microscope (SP8; Leica Microsystems) with Plan Apofluar 63 × 1.4 NA objective or a fluorescent microscope (DMi8; Leica Microsystems) with Plan Apofluar 63 × 1.4 NA objective using a digital CMOS camera (Hamamatsu Orca Flash 4.0 V2 Camera). Images were processed using Photoshop (Adobe) or ImageJ (National Institutes of Health, Bethesda, MD).

Quantitative immunofluorescence of centrosomal and ciliary levels of proteins was performed on cells by acquiring a z-stack of control, knockout, and rescue cells using identical gain and exposure settings, determined by adjusting settings based on the fluorescence signal in the control cells. The z-stacks were used to assemble maximum-intensity projections. The centrosome regions in these images were defined by γ-tubulin staining for each cell, and the total pixel intensity of a circular 3 μm² area centered on the centrosome in each cell was measured using ImageJ and defined as the centrosomal intensity. The ciliary regions in these images were defined by Arl13b or acetylated tubulin for each cell. Background subtraction was performed by quantifying fluorescence intensity of a region of equal dimensions in the area neighboring the centrosome or cilium. Ciliary protein concentrations were determined by dividing fluorescence signal of the protein to the cilium length, which was

quantified using Arl13b or acetylated tubulin staining. Primary cilium formation was assessed by counting the total number of cells and the number of cells with primary cilia, as detected by Arl13b or acetylated tubulin staining and DAPI staining. All data acquisition was done in a blinded manner.

Quantitative analysis of ciliogenesis efficiency in 3D cultures was determined by acquiring a z-stack of control, knockout, and rescue cells and assembling maximum-intensity projections. The number of cilia counted by Arl13b staining was divided into the number of cells counted by DAPI staining to calculate ciliogenesis efficiency of each spheroid and cell cluster.

### Fluorescence recovery after photobleaching analysis

Cells were cultured in coverglass-bottom dishes (Laboratory-Tek II Chamber) and serum-starved for 24 h. Photobleaching experiments were performed using Leica SP8 confocal laser scanner (Leica SP8) a Plan Apofluar 63 × 1.4 NA objective and a scan zoom of 5. Cells were kept at 37°C with 5% $CO_2$ during the imaging. A z-stack of 4 µm with 0.5 µm step size was taken during pre- and post-bleaching cilium FRAP experiments. One stack of images was captured before photobleaching, followed by photobleaching done in five iterations with 488-nm Argon laser with 100% power in the defined region of interest (ROI). Post-bleaching stacks were captured every 5 s for SSTR3-LAP and 6 s for HTR6-LAP for 2 min for half-cilium FRAP experiments and 5 min for full-cilium FRAP experiments. The relative fluorescence intensities were calculated from the maximum projection files using ImageJ and plotted against time. Recovery graph quantifications, t-half, and mobile pool quantifications were done using the equations as described [87].

### Transcriptome analysis

Transcriptome analysis was performed as described previously with the following modifications. Quality of reads of raw data was analyzed via FastQC program. We removed the sequencing reads with adaptor contamination and low-quality bases (quality score < Q30) via Trimmomatic (v0.35) to clean and advance the quality of the reads. All sequence data were 2 × 100 bp in length. The high-quality reads were saved in fastq files and deposited to NCBI's Gene Expression Omnibus under GEO Series accession number GSE123017. Clean and high-quality data were utilized for all downstream analysis. We used STAR aligner (v2.5.3) to generate genome indexes and mapping reads to the reference genomes that are hg38 for human cell line samples and GRCm38.p5 for mouse cell line samples. At the mapping step, reads having the non-canonical junctions were removed and only uniquely mapping reads were utilized. Following the alignment, Pearson's correlation analysis was conducted to obtain the transcript-level $R^2$ value between replicates. To determine the differentially expressed genes (DEGs), we utilized Cuffdiff with a minimum alignment count of 10, false discovery rate (FDR) < 0.05. The absolute value of |Fold change| ≥ 2 and *q*-value < 0.05 was used as the threshold to judge significant differences in gene expression. DAVID Bioinformatics Resources 6.8 [88,89] was utilized to determine the statistical enrichment of the significantly (*P* < 0.05) differentially expressed genes in KEGG pathways and GO-enrichment analysis. Analysis of GO annotation pathways provides information based on the biological processes (BP), cellular components (CC), and molecular functions (MF).

### Mass spectrometry

For the quantitative comparison of WT and knockout cells, snap-frozen cell pellets of $1 \times 10^6$ cells were resuspended in 50 µl PBS followed by the addition of 50 µl of 1% SDS in 100 mM HEPES, pH 8.4 and protease inhibitor cocktail (Roche, #11873580001). Samples were heated to 95°C for 5 min and then transferred on ice. Samples were treated with benzonase (EMD Millipore Corp., #71206-3) at 37°C for 1 h to degrade DNA. Protein concentrations were determined and adjusted to 1 µg/µl. 20 µg thereof were subjected to an in-solution tryptic digest using a modified version of the single-pot solid-phase-enhanced sample preparation (SP3) protocol [90,91]. Here, lysates were added to Sera-Mag Beads (Thermo Scientific, #4515-2105-050250, 6515-2105-050250) in 10 µl 15% formic acid and 30 µl of ethanol. Binding of proteins was achieved by shaking for 15 min at room temperature. SDS was removed by 4 subsequent washes with 200 µl of 70% ethanol. Proteins were digested with 0.4 µg of sequencing grade modified trypsin (Promega, #V5111) in 40 µl HEPES/NaOH, pH 8.4 in the presence of 1.25 mM TCEP and 5 mM chloroacetamide (Sigma-Aldrich, #C0267) overnight at room temperature. Beads were separated, washed with 10 µl of an aqueous solution of 2% DMSO, and the combined eluates were dried down. Peptides were reconstituted in 10 µl of $H_2O$ and reacted with 80 µg of TMT10plex (Thermo Scientific, #90111) label reagent dissolved in 4 µl of acetonitrile for 1 h at room temperature [92]. Excess TMT reagent was quenched by the addition of 4 µl of an aqueous solution of 5% hydroxylamine (Sigma, 438227). Peptides were mixed to achieve a 1:1 ratio across all TMT channels. Mixed peptides were subjected to a reverse phase clean-up step (OASIS HLB 96-well µElution Plate, Waters #186001828BA) and subjected to an off-line fractionation under high pH condition [90]. The resulting 12 fractions were then analyzed by LC-MS/MS on a Q Exactive Plus (Thermo Scientific) as previously described (PMID: 29706546). Briefly, peptides were separated using an UltiMate 3000 RSLC (Thermo Scientific) equipped with a trapping cartridge (Precolumn; C18 PepMap 100, 5 lm, 300 lm i.d. × 5 mm, 100 A°) and an analytical column (Waters nanoEase HSS C18 T3, 75 lm × 25 cm, 1.8 lm, 100 A°). Solvent A: aqueous 0.1% formic acid; Solvent B: 0.1% formic acid in acetonitrile (all solvents were of LC-MS grade). Peptides were loaded on the trapping cartridge using solvent A for 3 min with a flow of 30 µl/min. Peptides were separated on the analytical column with a constant flow of 0.3 µl/min applying a 2-h gradient of 2–28% of solvent B in A, followed by an increase to 40% B. Peptides were directly analyzed in positive ion mode applying with a spray voltage of 2.3 kV and a capillary temperature of 320°C using a Nanospray-Flex ion source and a Pico-Tip Emitter 360 lm OD × 20 lm ID; 10 lm tip (New Objective). MS spectra with a mass range of 375–1,200 m/z were acquired in profile mode using a resolution of 70,000 [maximum fill time of 250 ms or a maximum of 3e6 ions (automatic gain control, AGC)]. Fragmentation was triggered for the top 10 peaks with charge 2–4 on the MS scan (data-dependent acquisition) with a 30-s dynamic exclusion window (normalized collision energy was 32). Precursors were isolated with a 0.7 m/z window, and MS/MS spectra were acquired in profile mode

with a resolution of 35,000 (maximum fill time of 120 ms or an AGC target of 2e5 ions).

Acquired data were analyzed using IsobarQuant [93] and Mascot V2.4 (Matrix Science) using a reverse UniProt FASTA Mus musculus database (UP000000589) including common contaminants. The following modifications were taken into account: Carbamidomethyl (C, fixed), TMT10plex (K, fixed), Acetyl (N-term, variable), Oxidation (M, variable), and TMT10plex (N-term, variable). The mass error tolerance for full-scan MS spectra was set to 10 ppm and for MS/MS spectra to 0.02 Da. A maximum of 2 missed cleavages were allowed. A minimum of 2 unique peptides with a peptide length of at least seven amino acids and a false discovery rate below 0.01 were required on the peptide and protein levels [94].

### Mass spectrometry data analysis

The proteins.txt output file of IsobarQuant was analyzed using the R programming language. As a quality control filter, we only allowed proteins that were quantified with at least two different unique peptides. The "signal_sum" columns were used and annotated to the knockout (KO) and wild-type (WT) conditions. Batch effects were removed using the limma package, and the resulting data were normalized with the vsn package (variance stabilization) [90]. Limma was employed again to look for differentially expressed proteins between KO and WT. We classified proteins as hits with a false discovery rate (FDR) smaller 5% and a fold change of at least twofold. Candidates were defined with an FDR smaller 20% and a fold change of at least 50%.

### Statistical analysis

Statistical significance and $P$-values were assessed by analysis of variance and Student's $t$-tests using Prism software (GraphPad Software, La Jolla, CA). Error bars reflect SD. Following key is followed for asterisk placeholders for $P$-values in the figures: $****P < 0.001$ $***P < 0.001$, $**P < 0.01$, $*P < 0.05$.

Expanded View for this article is available online.

### Acknowledgements

We acknowledge the Firat-Karalar laboratory members, Moe Mahjoub, and Fanni Gergely for insightful discussions regarding this work. In particular, we thank Melis Dilara Arslanhan for support during network analysis of the global proteomics data. IMCD3 cells were a kind gift from Moe Mahjoub (University of Washington, St. Louis). We thank the European Molecular Biology Laboratory (Heidelberg) for the use of the Proteomics Core Facility for the TMT-based global proteome identification experiments. We thank the grants office in Cambridge Research UK for the management of the Newton Advanced Fellowship. This work was supported by ERC Grant 679140, EMBO Installation Grant, Newton Advanced Fellowship RG84475 managed by Cancer Research UK, and TUBITAK Grant 115Z521 to ENF. We also thank the Science Academy of Turkey for the BAGEP research award to ENF.

### Author contributions

EO and ENF-K conceived the project and designed the experiments. EO conducted and analyzed all of the experiments except for the RNAseq analysis. SG and IHK performed and analyzed the RNAseq experiments. EO prepared the figures, and ENF-K wrote the paper.

### Conflict of interest

The authors declare that they have no conflict of interest.

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
