## [Review Process File · EMBO Reports]

Centriolar satellites are required for efficient ciliogenesis and ciliary content regulation

Ezgi Odabasi, Seref Gul, Ibrahim H. Kavakli and Elif N. Firat-Karalar.

Review timeline:

Submission date:	14 th January 2019
Editorial Decision:	15 th January 2019
Revision received:	27 th February 2019
Editorial Decision:	12 th March 2019
Revision received:	21 st March 2019
Accepted:	1 st April 2019

Editor: Deniz Senyilmaz Tiebe

Transaction Report:

1st Editorial Decision

15th January 2019

Thank you for submitting your manuscript entitled " Differential requirements for centriolar satellites in cilium formation across different cell types" to EMBO Reports. Your manuscript was previously reviewed at another journal, and you provided a preliminary point-by-point response to those referee comments. I have looked at everything and I believe the proposed revision will strengthen the manuscript.

Therefore I would like to invite you to revise your manuscript with the understanding that the referee comments must be fully addressed and their suggestions taken on board. Please address all referee concerns in a complete point-by-point response. Acceptance of the manuscript will depend on a positive outcome of the review. It is EMBO Reports policy to allow a single round of revision only and acceptance or rejection of the manuscript will therefore depend on the completeness of your responses included in the next, final version of the manuscript.

Referee #1:

The centriolar satellites are poorly-understood regulators of centriole dynamics and cilium formation; they are also believed to have additional regulatory functions in cells. Odabasi, Firat-Karalar and colleagues here explore the roles of the centriolar satellite protein, PCM1, in controlling primary ciliogenesis in cultured cells. They present a phenotypic comparison between PCM1-edited hTERT-RPE1 cells (human, retinal pigmented epithelial) and IMCD-3 cells (murine, kidney, SV40-transformed). A previously-published study has described in detail the PCM1-deficient phenotype seen in hTERT-RPE1 cells (Wang et al. (2016) eLife 5:e12950), using a very similar strategy to that used here to ablate PCM1. The current submission confirms the principal findings of this paper with respect to the ciliogenesis defect in hTERT-RPE1 cells. The phenotype of defective ciliogenesis seen in IMCD-3 cells is qualitatively similar to that seen in hTERT-RPE1 cells, although less pronounced, and some specific centriolar/ciliary proteins are affected differentially between the two cell lines.

The juxtaposition of human and mouse cells from different tissues, with different immortalisation status and ciliogenesis mechanisms, does not allow the authors to define a clear function for the centriolar satellites. Although several potentially-important differences are observed between PCM-deficient hTERT-RPE1 and IMCD-3 cells, it is not clear whether any one of these differences is of pivotal importance in the contribution made by satellites to regulating primary cilium formation. The experiments are robust and generally well described, but the extent to which the observations presented here offer a mechanistic advance over previous work is limited.

We thank the reviewer for the accurate summary of our findings and for the constructive criticism of our manuscript. We are very happy to see the reviewer found the data presented in the manuscript as robust and well described. We agree with the reviewer's concern on the limitations of using human RPE1 and mouse IMCD3 cells to compare phenotypic and molecular consequences of satellite loss. Given that the RPE1::PCM1^{-/-} characterization for ciliogenesis defects was already published, as pointed out by the reviewer, we removed the RPE1-related data on cilium assembly and global transcriptomics/proteomics from the manuscript along with our conclusions on cell type specific differences. Instead, we rewrote the manuscript to emphasize our findings from kidney epithelial cells on direct satellite functions in cilium- and centrosome-related functions and mechanisms, as addressed in detail below in our response.

Regarding the criticism of this reviewer on the lack of mechanistic advance of our study, we disagree, and argue that our work does provide a significant advance in our understanding of the centriolar satellites function and mechanism in two major ways, which we emphasized in the revised manuscript:

- As reported by Wang et al. 2015 eLife:e12950 and our study, RPE1 PCM1^{-/-} cells did not ciliate and thus they did not allow addressing the function of satellites in cilium-related functions. Since IMCD3 PCM1^{-/-} cells ciliated with a reduced efficiency but still formed full-length cilia, this system allowed us to address previously uncharacterized ciliary functions of satellites, which are among the predominant defects underlying ciliopathies. In this study, we for the first time identified direct roles for satellites in regulation of ciliary content, timely response to Hedgehog signals and epithelial cell organization. However, we did not identify functions outside the cilium context including cell cycle progression, cell proliferation and centriole duplication both in IMCD3 and RPE1 cells, all of which were not addressed in the Wang et al. paper. Additionally, at the mechanistic level, we discovered that the ciliogenesis defects in IMCD3::PCM1^{-/-} cells was not due to increased Mib1 accumulation at the centrosome in contrast to RPE1::PCM1^{-/-} cells. Instead we identified the defects in basal body IFT recruitment as the likely underlying mechanism for the ciliogenesis defect in these cells.
- I would like to highlight that centriolar satellites is the third component of the mammalian centrosome/cilium complex. Despite the lack of direct mechanistic insight into all satellite-linked functions we reported in the manuscript, our study will be a major advance in our understanding of which centrosome/cilium complex-related functions and mechanisms are specifically regulated by centriolar satellites. The contribution will be analogous to the way Sir et al. 2013 JCB: 201309038 study advanced how we think about centrioles through phenotypic characterization of centriole-less vertebrate cells.

1. A concern is the presence of a second band in the PCM1 immunoblot in IMCD3 cells, allied with the detection of PCM1 peptides in the KO samples in the mass spectrometry analysis in Table S2 (line 2967). Is there any possibility of a cryptic start site that may result in a truncated PCM1 protein that adopts a non-satellite localization, but that can support limited primary ciliogenesis? Does an immunoblot with the C-terminal antibody used in Figure 1 C address this issue? The authors should discuss this issue more specifically as it could impact significantly on their findings.

We agree with the reviewer's concern. Although the immunofluorescence with antibodies targeting N-terminal and C-terminal antibodies confirmed lack of PCM1 signal, the presence of an extra band that we defined as nonspecific is concerning. We addressed this issue in two ways:

- The polyclonal C-terminal antibody we generated in the lab did not detect mouse PCM1 in immunoblotting experiments. Therefore, we ordered three different polyclonal antibodies targeting near C-terminus of PCM1. Only one of them, raised against the 630-726 amino acids of PCM1, detected

mouse PCM1 in immunoblotting experiments. Immunoblotting of the cell extracts from control cells and the three IMCD3 PCM1 KO clones with this antibody confirmed lack of PCM1 expression in KO cells (Fig. 1A).

- We discussed the TMT-labeling data for control and PCM1 KO cells with our collaborators at EMBL Proteomics Facility. They told us that given the nature of TMT-labeling experiments where control and KO samples were mixed and due to an effect known as “ratio compression”, the complete lack of signal for PCM1 KO cells were not expected and that the more than 4 fold reduction of PCM1 in KO cells is in the range of what is commonly observed with this quantitative mass spectrometric approach for proteins which have been knocked out.

2. The authors should test whether the changes in centrosomal/ basal body proteins seen in Figure 4 are affected by serum starvation, i.e. under ciliogenesis conditions.

We agree that quantification of signal under serum starvation is physiologically more relevant to study the molecular basis of ciliogenesis defects. As suggested by the reviewer, we performed quantification of centrosomal levels of key ciliogenesis factors in cells serum starved for 24 hours in addition to the ones we performed in asynchronous cells. This analysis showed that the changes in the centrosomal levels of the ciliogenesis factors under serum starvation were similar to the changes in asynchronous cells. We included the quantification data from serum-starved cells to Fig. 4 and moved the data for asynchronous cells to Fig. S4 Finally, we detailed the conditions for different quantifications in the related text and figure legends.

3. Can it be determined whether IFT rates are altered in the absence of PCM1?

We were not able to generate a stable line expressing IFT components in the PCM1 KO cells to perform these experiments. Given that the ciliary levels of SSTR3 and HTR3 were also significantly reduced in satellite-less cells like IFT88, we addressed the reviewer’s point on the possible function of satellites in ciliary dynamics of proteins by performing half and full cilium experiments for HTR6 and SSTR3 in control and PCM1 KO cells, using the protocols developed by the Hu, Milenkovic et al. 2010 PMID: 3092790 study. Half cilium FRAP experiments of HTR6-LAP and SSTR3-LAP showed that the percentage and halftime of recovery were similar between control and PCM1 KO cells (Fig. S6B-D). Moreover, full cilium FRAP experiments showed no recovery at the cilium for both control and PCM1 KO cells (Fig. S6A). Together, these results show that satellites are required for regulating the ciliary levels, but not dynamics, of HTR6 and SSTR3.

4. A rescue experiment should be included for the SAG experiment in Figure 6 and for the spheroid experiment in Figure 7.

- To validate that the defects we reported for Smo ciliary relocalization upon SAG treatment was specific to loss of satellites, we performed rescue experiments using the LAP-PCM1::KO cells. Reduction in Smo relocalization phenotype was rescued by stable expression of LAP-tagged full-length human PCM1 at 4h and 8h time points of SAG stimulation (Fig. 6A).

- As for the rescue experiments for spheroid assays, we attempted these experiments with the LAP-PCM1::KO cells, however they were inconclusive due to very low spheroid formation efficiency of the control cell line itself. The decrease in the spheroid formation efficiency was likely because of the second round of single cell cloning protocol for generating Flip-In stables on top of generating CRISPR KO clones. In our experience, the spheroid formation efficiency of these cells is sensitive to multiple rounds of single cell cloning. As an alternative approach, we performed the spheroid rescue experiments with cells transiently transfected with myc-PCM1 according to the previously published protocols (Slaats et al. 2016 PMID: 26490104). Quantification of these experiments showed a partial but significant rescue of spheroid formation (Fig 7A and 7B). Of note, we used myc-PCM1 instead of LAP-PCM1 for rescue experiments because we are limited in using four colors for staining (Arl13b, beta-catenin, ZO1, DAPI). We included immunofluorescence data in Fig. S7 confirming satellite localization of myc-PCM1 fusion.

5. One cannot know whether the different effects seen of PCM1 editing in the hTERT-RPE1 cells versus the IMCD-3 cells are due to species-specific differences, transformation status or to the tissue of origin of the cells, without controlling for these variables. There appears to be a notable difference in the relative levels of PCM1 (total or in satellite) between the cells studied and thus the relative impact of satellite disruption may vary. Therefore, the extent to which this study defines a differential requirement for centriolar satellites between different cells is questionable.

We fully agree with the reviewer on limitations of using two different cell lines of different origin and immortalization status in deriving conclusions on cell-type specific differences. As the reviewer pointed out earlier, Wang et al. 2015 eLife:e12950 paper reported that characterization of RPE1::PCM1-/- cells for inhibition of primary cilium assembly. Because RPE1 ciliogenesis-related data was already published and the major advance of our paper is the extensive molecular and phenotypic characterization of kidney epithelial cells for previously undescribed processes, we revised the manuscript to focus on these findings and removed our conclusions on cell-type specific differences for satellite functions and mechanisms. Finally, we added the following sentence to discussion to explain the differences in the ciliogenesis phenotypes in satellite-less RPE1 and IMCD3 cells in pg. 16 “The variation in the phenotypes observed in IMCD3 and RPE1 cells might be due to differences in the species and tissues they are derived from, ciliogenesis mechanisms and mechanism of transformation.”

6. Comparison of the proteomic alterations between cells is potentially informative, but the findings here do not provide a strong model for how satellites contribute to ciliogenesis. A comparison of the proteomes under conditions of serum starvation would be of interest, but the divergence in the proteins affected under asynchronous conditions suggests this may be speculative.

The motivation of performing these proteomic studies was 1) testing the role of satellites in proteostatic regulation of centrosome proteins, which was suggested by previous studies including the Wang et al. 2015 eLife:e12950, but was done in a piecemeal way for a subset of proteins 2) gaining unbiased insight into satellite-linked functions outside the centrosome/cilium context. Surprisingly, we found that the centrosome proteome was unaltered in IMCD3 PCM1 KO cells and yet pathways like cell migration and adhesion were altered. Although the proteomics work do not contribute to our understanding of ciliogenesis pathways, it is significant in addressing the two motivating questions we started with and thus provides an important resource for the field. We revised the part of the manuscript relevant to this data to emphasize our motivating questions and unexpected findings from this systematic analysis.

Minor points

7. It is unclear why 'vertebrate' is used instead of the more specific 'mammalian' in the title.

We changed the title to “Centriolar satellites are required for efficient ciliogenesis and ciliary content regulation”.

8. The sourcing of the hTERT-RPE1 cells should be clarified.

hTERT-RPE1 cells were purchased from ATCC (ATCC® CRL-4000). We included this information in “Materials and Methods” part.

9. Details of the genome editing should be provided more clearly, enabling the reader to follow where in the mouse and human coding sequences has been disrupted and whether this is comparable to the strategy used in the Wang et al. paper in eLife. Supplementary Figure 1 indicates translation of exon 4, which was apparently not targeted, which is confusing. Using a font with equal character spacing would enable the indels in the different alleles to be identified more easily.

The gRNA used in the Wang et al. paper targeted “coding exon 1” that corresponds to “exon 3” of human PCM1. The gRNA we used for RPE1 cells targeted “coding exon 2” that corresponds to “exon 4” of human PCM1. The gRNA we used for IMCD3 cells targeted “coding exon 2” that corresponds to “exon 3” of mouse PCM1. We included these details in figure legends. As

suggested by the reviewer, we also reorganized Fig. 1 (now Fig. S1) to align the two alleles by using fonts of equal character and also included a legend annotating the different type of mutations.

10. Size markers should be included in all the immunoblots.

We included size markers for all the immunoblots.

11. Blow-up panels would be helpful for Figure 1C and Supplementary Figure 1D and F.

We included blow-up panels for the suggested figures.

12. Figure 2B should distinguish the 2 sets of data (one presumes this is different timing of serum starvation, but this should be clarified).

We now included the timing of serum starvation associated with the two different data points.

13. Labelling of the different parts of Figure 3 should be revised in the relevant Legend.

We corrected the labeling of Figure 3 as suggested.

14. The x-axis in Figure 3A should have the correct numerical intervals, not categories.

We performed the proliferation assays by counting cells in days 1,2, 3 and 5, as represented in Fig. 3A.

15. Representative FACS plots should be shown to support the data in Figure 3B.

We included the representative FACS plots in Fig. S2 for IMCD3 cells and Fig. S3G for RPE1 cells

16. It is unclear how the data in Figure 3E are supported, as the large number of G1 cells in the asynchronous population should have single centrosomes (with 2 centrioles), but the G2 and M cells should have 2 (with 4 centrioles). This should be clarified. Centriole counts would be more informative than 'centrosomes'.

We agree with the reviewer that centriole counts will be more informative and performed perform these experiments in cells stained for the centriole marker "centrin 2 and 3". These counts did not reveal a significant defect in centriole duplication, and we included the representative data and quantification in Fig. 3E.

17. How the ciliary 'concentrations' were determined should be defined in individual experiments, i.e., on the basis of Arl13B length or of acetylated tubulin. It is unclear from the plots presented in Figure 5 that 100 data points are included in each analysis. It would be appropriate not to present all of the data points with that number, but the small number included for Sstr3 and Htr6 is confusing.

- We determined the ciliary concentrations by dividing the ciliary signal for the protein of interest to the cilium length, which was quantified based on staining with antibodies against acetylated tubulin or Arl13B staining.

- As for the plots presented in Fig. 5 and Fig. S5, we performed each experiment two independent times and normalized the data points of each experiment to the mean of that experiment (=1) in order to account for differences in the fluorescence intensities between different experiments. In particular for LAP-SSTR3 and LAP-HTR6, we quantified more cells and included more data points in Fig. 5B and 5C.

Referee #2:

The manuscript by Odabasi et al deals with the physiological roles of centriolar satellites (CS) in mammalian cells. To this end, the authors delete the gene encoding PCM1 in human RPE1 cells and murine IMCD3 cells, a critical scaffold for CS that is not known to play independent roles directly at the centrosomes. The authors build on a previous study showing an absolute requirement of PCM1 for cilium formation in RPE1 cells. For IMCD3 cells, which differ from RPE1 cells in their requirements for cilium formation, this dependency on PCM1 is only partial. Instead, PCM1 deficiency in this background only leads to ciliation defect in 50% of the cells, but is associated with defective hedgehog signaling originating from the cilium. The authors also show that lack of CS in IMCD3 cells is not associated with differences in basic centrosome characteristics, cell cycle progression or mitosis.

Despite an affluence of papers on CS and their postulated involvement in diverse cellular processes, there is only little actual evidence for many of these claims. Especially the requirement of CS outside cilium formation warrants proper characterization. Thus, the manuscript is both important and timely. Despite the lack of mechanistic insight into the differential requirements of CS for cilium formation in diverse cellular backgrounds, this work goes a long way to clarify the key functions of CS. The data are in general of a good quality, and conclusions are not overstated. I do not feel that the transcriptomic and proteomic studies add much to the story. Rather, I would have liked to see the authors explore some of the basic observations of CS' roles in the RPE1 system as well. Despite the authors have managed to produce a rescued cell line, critical rescue experiments are lacking in most places. Finally, it is unclear whether the cilium-associated signaling defects are due to the reduction in ciliated cells or represents a defect

in the cilia formed.

We thank the reviewer for the positive assessment and constructive feedback of our manuscript.

1- Figure 3: These data very convincingly show that CS (at least when permanently removed) does not affect cell proliferation, cell cycle distribution, centrosome numbers or cell division in IMCD3 cells. These data are very clear and credible, but at odds with various other papers (using less convincing methods). I think it would be highly beneficial to extend these studies to the RPE1 background.

We agree with the reviewer on the importance of negative data related to centrosome-associated functions for satellites in a constitutive knockout background. To corroborate these findings and to address the generality of these phenotypes, we performed the same experiments in control and RPE1 PCM1 KO cells. As now included in Fig. S2F, S2G and S2H, we did not observe any significant defects in cell cycle progression, cell proliferation and centriole duplication in RPE1 PCM1 KO cells relative to control cells.

2- Figure 4: There is something odd with these stainings. I wonder why the authors only detect proteins like CEP290 and MIB1 at centrosomes and not at CS, where they should normally be more enriched than on the actual centrosomes. Also, the IFT88 stainings clearly indicate that cilium formation has been induced, but there is no mention of this in the text or in the figure legends.

- As the reviewer pointed out, we observed a more centrosome-restricted Cep290 and Mib1 localization in IMCD3 cells. When we used the same antibodies in RPE1 cells, we observed prominent satellite localization in agreement with the literature. Therefore, we believe that this difference is likely due to variation in the satellite distribution of these proteins in different cell types.
- We now have two figures on quantification of centrosomal levels of key ciliogenesis factors: Fig. 4 for serum-starved cells and Fig. S4 for asynchronous cells. For both conditions, we observed similar changes in the centrosomal levels of these proteins. To clarify these points, we included more detail on the conditions we used for quantification in figure legends and related text in the manuscript.

3- Figure 4+5: There are no specificity controls for the antibody stainings, which would have raised the confidence of the experiments. Given the sometimes small (but significant) differences between the WT and PCM1 KO conditions, rescue by reintroduction of ectopic protein would raise the credibility of the data. The authors have already managed to make such a rescue cell line with a BAC.

We agree with the reviewer's concern regarding antibody specificity. To address this, using the LaP-PCM1 IMCD3 KO rescue line, we performed the

quantifications in Fig. 4 and Fig. 5 for all proteins except for LAP-SSTR3 and LAP-HTR6. We could not perform these quantifications for SSTR3-LAP and HTR6-LAP as these are both Flippin stable lines and therefore could not be generated in the LAP-PCM1::KO parental line. Stable expression of LAP-PCM1 rescued the changes we observed for all proteins including Ift88, Cep164, Mib1 and Talpid3, confirming the specificity of these phenotypes. Of note, we would like to emphasize that all the antibodies we used in this study were validated in previous papers and we now included the catalog numbers/source information for all antibodies in the “Materials and Methods” section.

4- Figure 6: The defects observed in the hedgehog signaling pathway are very interesting and hints to a direct physiological consequence of compromised ciliogenesis in IMCD3 cells. Yet it remains unclear to me whether this signaling defect is simply the result of less cilia being formed or that those that do form are still defect. After all, the effect size is rather similar (Fig. 2B compared to Fig. 6A). I think this is an important point that needs clarification. Especially for the experiments in this figure, inclusion of rescue conditions are important.

- To validate that the defects we reported for Smo ciliary relocalization upon SAG treatment was specific to loss of satellites, we performed rescue experiments using the LAP-PCM1::KO cells. Reduction in Smo relocalization phenotype was rescued by stable expression of LAP-tagged full-length human PCM1 at 4h and 8h time points of SAG stimulation (Fig. 6A).

- We agree with the reviewer’s concern on whether Hedgehog defect is a direct consequence of satellite loss or not. We performed quantifications of ciliary Smo levels in the cilia that formed in IMCD3 PCM1 KO and control cells. We added the following sentence in pg 12 “In the IMCD3 PCM1 KO and control cells that formed cilia, we determined ciliary Smo levels...” to include more detail on quantification. Therefore, we can conclude that the reported defects are not due to reduction in ciliation efficiency. Although the cilia that formed in IMCD3 PCM1 KO cells were of similar length to control cells, it is possible that the ones that formed were not mature or functional enough. To include this possibility, we added the following sentence in pg 12 “ ... lack of satellites directly or indirectly caused a delay in translocation of Smo to the cilium in response to SAG (Fig. 6A).”

5- Figure 7: these 3D culture experiments are very nice indeed, but I wonder why the authors didn't also look for a ciliogenesis defect under these more physiologically relevant growth condition. Based on the Arl13b stainings in Fig. 7A there is a clear defect that should be quantified, and rescue experiments would again be beneficial.

- To quantify the percentage of ciliogenesis in the 3D spheroids, we stained the 3D cultures of control and PCM1 KO cells with the ciliary marker Arl13b and the DNA stain “DAPI”, acquired a z-stack of the clusters and spheroids that were chosen in a blinded manner and counted the number of cilia and cells using the

maximum-intensity projections. Ratio of cilia to cells were presented as the “ciliogenesis efficiency per spheroid” in Fig 7C. This analysis revealed a significant decrease in the ciliation efficiency of PCM1 KO cells relative to control cells, in agreement with the results we obtained in 2D cultures.

- As for the rescue experiments for spheroid assays, we attempted these experiments with the LAP-PCM1::KO cells, however they were inconclusive due to very low spheroid formation efficiency of the control cell line itself. The decrease in the spheroid formation efficiency was likely because of the second round of single cell cloning protocol for generating Flip-In stables on top of generating CRISPR KO clones. In our experience, the spheroid formation efficiency of these cells is sensitive to multiple rounds of single cell cloning. As an alternative approach, we performed the spheroid rescue experiments with cells transiently transfected with myc-PCM1 according to the previously published protocols (Slaats et al. 2016 PMID: 26490104). Quantification of these experiments showed a partial but significant rescue of spheroid formation (Fig 7A and 7B). Of note, we used myc-PCM1 instead of LAP-PCM1 for rescue experiments because we are limited in using four colors for staining (Arl13b, beta-catenin, ZO1, DAPI). We included immunofluorescence data in Fig. S7 confirming satellite localization of myc-PCM1 fusion.

6- Figure 8, Table 1. I fail to see how these experiments add to the story.

The motivation of performing these proteomic studies was 1) testing the role of satellites in proteostatic regulation of centrosome proteins, which was suggested by previous studies including the Wang et al. 2015 eLife:e12950 but was done in a piecemeal way for a subset of proteins 2) gaining unbiased insight into satellite-linked functions outside the centrosome/cilium context. Surprisingly, we found that the centrosome proteome remained mostly unaltered in PCM1 KO cells, which we confirmed for a subset of centrosome/cilium proteins in Fig. 4. However, the pathways like cell migration and adhesion were altered. Although the proteomics work do not contribute to our understanding of ciliogenesis pathways, it is significant in addressing the two motivating questions we started with and thus provides an important resource for future studies that aims at studying satellite functions and mechanisms in other contexts. We revised the part of the manuscript relevant to this data in order to emphasize our motivating questions and unexpected findings from this systematic analysis.

Referee #3:

Odabasi et al. examine the effects of removing a centriolar satellite protein in IMCD3 cells, and somewhat in RPE1 cells. In many ways, this work builds on Wang et al. eLife 2016, which examined CRISPR-mediated deletion of PCM1 in RPE1 cells and many of the same proteins examined by Odabasi et al. The authors confirm that deletion of PCM1 disrupts centriolar satellites, and that these cells have decreased cilia biogenesis. Removal of PCM1 does not affect

cell proliferation or the duration of mitosis or centrosome number. Immunofluorescence suggests that there is a reduction in IFT88 and HTR6. Loss of PCM1 also has a transient effect on SMO accumulation in cilia, and may have an effect on Hedgehog signaling. There is a diminishment of spheroid formation, similar to other genetic perturbations of ciliogenesis. The authors also apply transcriptomics and proteomics. There are few changes to the transcriptome, which are not examined more. There are extensive small changes to the proteome, again the significance of which is not tested.

Most of the data in this manuscript seem to be of good quality and competently generated. However, the work suffers from some overinterpretation, and one extremely egregious overinterpretation. The title, abstract and much of the discussion focus on differential requirements for satellites in different vertebrate cell types. Indeed, when I read in the abstract that, "While satellites were essential for cilium assembly in retinal epithelial cells, kidney epithelial cells lacking satellites still formed full-length cilia," I had high expectations for an interesting examination of tissue-specific functions of centriolar satellites. However, this paper is almost entirely about loss of PCM1 in the IMCD3 cell line, with some scanty data about loss of PCM1 in the RPE1 cell line. While the RPE1 line is derived from retinal pigmented epithelium, and the IMCD3 cell line is derived from kidney, neither should be taken as faithful recapitulations of the biology of their tissues of origin.

Moreover, the authors conclude that they have identified "cell type-specific roles for satellites and provide insight into the phenotypic heterogeneity of ciliopathies." Even if we were to say that these cell lines derived over two decades ago faithfully recapitulate the biology of their tissues of origin, are the differences they observe due to the different cell types, or are they due to the different species (IMCD3 cells are mouse, RPE1 cells are human), or due to the different mechanisms of transformation (IMCD3 cells are immortalized with SV40, RPE1 cells with hTERT), or other idiosyncratic aspects of each cell line? None of these other variables are tested, or even mentioned. As these cell lines are very different, it is extremely problematic to make any conclusions about differences in phenotype, especially when the differences are in degree (the authors observe that the knockout RPE1 cells show a large decrease in cilium formation, whereas IMCD3 knockouts show a more moderate decrease). Moreover, there is no explanation as to why or insight into underlying mechanisms accounting for differences.

Somewhat perplexingly, the authors finish the manuscript with mass spectrometry-based analyses of PCM1 knockouts, and refer to the RPE1 data without actually including it in the manuscript other than a single summary figure (Figure 8D). This level of transparency falls far below standards in the field. For a paper that purports to be about comparisons, not providing the most unbiased data about that comparison is mystifying.

In summary, this work should not be published as the central conclusion is not justified by the data.

We thank the reviewer for the accurate summary of our findings and the constructive criticism of our manuscript. We are happy to see that the reviewer found the data presented in the manuscript as of good quality and competently generated. The major concern raised by this reviewer is related to our conclusions on cell-type-specific functions of centriolar satellites, which we proposed based on phenotypic and molecular comparison of human RPE1 and mouse IMCD3 cells. We fully agree with the reviewers' concerns on why use of two cell lines from different tissues and organisms and immortalization states is not sufficient to derive these conclusions and comes across as overinterpretation. However, possibly due to how we titled the manuscript and discussed some of our data, the major advances of our manuscript on characterization of satellite-specific functions and mechanisms, which are important and timely for the field, might have been underestimated.

To address the reviewers' major concern, we included in the manuscript only the data related to characterization of IMCD3 cells and rewrote the paper to emphasize the major conclusions on 1) identification of satellite-specific functions in ciliary content regulation, Hedgehog signaling and epithelial cell organization, but not in processes outside the cilium context including in centriole duplication, cell cycle progression and cell proliferation 2) molecular dissection of the ciliogenesis defects of IMCD3 KO cells, which revealed a defect in basal body IFT recruitment, but not in Mib1 accumulation 3) systematic analysis of the global proteome of satellite-less cells that identified centrosome proteome to remain as mostly unaltered and revealed changes in other pathways.

In the revised version, we removed the RPE1-related data on cilium assembly, global proteomics and transcriptomics, as well as our conclusions on cell-type specific differences. Prior to our work, the phenotypic characterization of RPE1 PCM1 KO cells during cilium assembly was already published in 2016 in eLife and thus RPE1-related data related to ciliogenesis do not contribute to our above-mentioned major conclusions. Finally, we added the following sentence to discussion to explain the differences in the ciliogenesis phenotypes in satellite-less RPE1 and IMCD3 cells in pg. 16 "The different phenotypes observed in IMCD3 and RPE1 cells might be due to differences in the species and tissues they are derived from and mechanism of transformation." Of note, upon suggestion of one of the reviewer's, we performed centriole duplication, cell proliferation and cell cycle progression experiments in control and RPE1 PCM1 KO cells, which was not published previously in the Wang et al. 2016 paper and included this data in Fig S3F, S3G and S3H.

1- The authors highlight in the abstract that loss of PCM1 affects "actin cytoskeleton pathways and neuronal functions." In fact, neither are tested. GO analyses of proteome differences are a starting point for more experimental

assessment, not conclusions in themselves.

Our motivations in performing these global proteomic studies in PCM1 KO cells were as follows:

1) We tested the role of satellites in proteostatic regulation of centrosome proteins, which was suggested by previous studies including the Wang et al. 2015 eLife:e12950. In contrast to the previous studies that in a piecemeal way suggested such regulation, our analysis showed that the centrosome proteome mostly remains unaltered in satellite-less cells and that satellites likely mediate their functions through regulation of protein targeting to the centrosomes and cilia, as supported in Figure 4 and 5.

2) We aimed to gain unbiased insight into satellite-linked functions outside the centrosome/cilium context. While we believe that these datasets provide a powerful resource for future studies aimed at studying satellite function and mechanism, we think that following up on possible function of satellites in actin cytoskeleton pathways and neuronal functions is beyond the scope of this paper, we. In order to avoid drawing conclusions on pathways we did not test in the context of this manuscript, we removed the “actin cytoskeleton pathways and neuronal functions” part from the abstract and included it as part of the discussion of the proteomics data.

Although we agree with the reviewer that most of the time proteomic analysis is the starting point of most papers, due to the above-mentioned reasons, we would like to keep these data following the functional dissection in the manuscript. We revised the part of the manuscript relevant to this data in order to emphasize our motivating questions and unexpected findings from this systematic analysis.

3- Figure 6D. Comparative amounts of Gli1 only in the presence of Gli1 do not seem very informative. The standard in the field seems to be to show Gli1 levels in both the absence and presence of agonist.

We agree, and thus performed the transcriptional assay in the presence and absence of the agonist and included this data in Fig. 6D and Fig. 6E.

4- The Methods section is insufficient. To improve reproducibility, methods should be detailed enough for others to reproduce the work without referring to other sources. For example, antibodies are lacking RRIDs.

We agree, and improved the methods section by including enough details on the experimental protocols in order to ensure reproducibility. In particular, we revised the following parts:

- We included information on RRIDs of antibodies and cited the papers that validated the home-made antibodies.
- We included detailed protocol on how we generated LAP-PCM1, HTR6-LAP and SSTR3-LAP stable cell lines using the Flip-in system.
- We included more detail on the immunofluorescence staining,

immunoblotting protocols and cloning.

- We included more details on how we performed quantification of centrosomal and ciliary levels and concentrations of proteins as well as ciliation efficiencies in 2D and 3D cultures.

5- As the authors note, epithelial spheroids have been widely used to assay cilia dysfunction. Given that the authors have documented cilia dysfunction in monolayers, what does the spheroid assay add to the work?

We do not agree with the reviewer on this point. The significance of the epithelial spheroid assays is not relevant to assaying defects in cilium assembly and function. Instead, they are powerful in determining the functional consequences of cilia dysfunction on tissue architecture. Therefore, they provide information on processes that we cannot study in 2D cultures and have been used widely in the field (Mahjoub and Stearns 2012 PMID: 22840514, Wheway et al. 2015 PMID: 4536769 etc...). Through using this *in vitro* tissue model, we showed that lack of satellites disrupted epithelial organization for the first time, which provides insight into the possible *in vivo* functions of satellites.

6- Why are different methodologies used for IMCD3 (TMT) and RPE1 (SILAC) cell mass spectrometry analyses? If the goal is to compare the effects in two different cell lines, then changing the methodology adds a major confounding factor. There are no experiments to validate the differences detected by the mass spectrometry.

- We agree with the reviewer that the SILAC data presented for RPE1 cells is of poor quality, should be done using the TMT-based approach to have consistency and must be improved. However, given that we removed RPE1-related ciliogenesis data from the revised version of the manuscript, we also decided to remove the transcriptomics and proteomics data for RPE1 PCM1 KO cells. Along the lines of the major criticism of this reviewer, comparing two proteomics data of two very different immortalized cell lines will not provide mechanistic insight into cell type specific differences and results in underestimation of our conclusions from the systematic analysis.

- As for the validation of the proteomics data for IMCD3 KO cells, we now included validation of the IMCD3 PCM1 KO proteomics data by performing immunoblotting for the centrosome proteins that are among the most upregulated and downregulated and for which we could purchase antibodies. In agreement with the proteomics data, we observed a significant increase in the cellular abundance of Cep131 and a significant decrease in the cellular abundance of SLAIN2 in PCM1 KO cells relative to control cells. We included this data in Fig 4G.

7- Figure 8C is misleading in that the insets have a different y-axis definition, and have no y-axis label to indicate that difference.

We agree, and now revised the y-axis of all three graphs to have the same interval and end value.

8- Much of the work could be moved to supplemental data, including confirmation of CRISPR knockout (Figure 1), negative data (Figure 3), confirmation of a cilia-dependent effect (Figure 7) and GO analysis (Figure 8).

We agree with the reviewer that part of Figure 1 related to the sequencing the clones can go to the supplement as well as the transcriptome data and GO analysis results for global proteome. However, we decided to keep the rest of the data as main figures because they support our conclusions on satellite-specific functions and mechanisms. Figure 3 for the first time reports that satellite-less kidney epithelial cells are not defective in cell proliferation, cell cycle progression and centriole duplication, processes outside the centrosome context. Figure 7 is important for reporting the consequences of satellite loss on epithelial organization.

Thank you for submitting the revised version of your manuscript. It has now been seen by two of the original referees.

As you can see, both referees find that the study is significantly improved during revision and recommend publication. However, the referees have some remaining minor concerns. In particular, referee #1 found that as it stands Fig. 4G does not clarify the relative increase in abundance of centriolar and total protein levels in PCM1null cells. Moreover, referee #1 questions whether induction of centriolar duplication has comparable effects on wild type and PCM1 null cells. Referee #2, on the other hand, would like to see additional controls regarding knockdown efficiency and levels of overexpressed proteins. I think it would be good to sort these out and I would like to discuss with you what could be done to address these comments in a reasonable timeframe. You might already have good arguments/data at hand regarding these points. Before you embark on the revisions please contact me to discuss this issue further.

Thank you again for giving us to consider your manuscript for EMBO Reports, I look forward to your minor revision.

REFeree REPORTS

Referee #1:

The key finding is the definition of the cilium-specific impacts of centriolar satellite ablation. This work is significant and extends our understanding of the roles of centriolar satellites and the mechanisms of ciliogenesis. This will be of general interest to the molecular biology community because centriolar satellites are poorly understood but of importance in a range of cellular activities. The principal finding of the study is robustly documented, using a range of different approaches.

Odabasi et al. here explore in detail the role of centriolar satellites in ciliary assembly and functioning by ablation of the key scaffold protein, PCM1, in murine IMCD3 kidney epithelial cells by CRISPR-mediated genome editing.

They demonstrate the loss of centriolar satellites resulting from PCM1 loss and a marked reduction of ciliogenesis in serum starved IMCD3 cells. A parallel experiment in human hTERT-RPE1 retinal pigmented epithelium cells showed an almost complete loss of ciliogenesis capacity, consistent with previously-published data in this cell line. They show no major impact of PCM1 ablation on cell cycle or centriole duplication. However, the authors show clearly that PCM1 loss affects the levels of key ciliary regulators in both serum starved and asynchronous cells. The intraflagellar transport complex B protein Ift88 levels declined at the basal body, whereas those of the appendage protein Cep164 increased. Notably, the basal body levels of Mib1 and Talpid3 were affected in a quantitatively different manner by PCM1 loss in IMCD3 cells compared with hTERT-RPE1 cells. Defective recruitment of ciliary proteins Ift88, Sstr3 and Htr6 was seen in the absence of PCM1 and a clear deficiency in the movement to cilia of Smo and Hh activation was also observed. In a 3D spheroid model of kidney epithelial cell assembly, PCM-deficient cells were notably defective and also showed a decline in ciliation. The authors then present a detailed analysis of the impact of PCM1 deficiency on the IMCD3 transcriptome and proteome, finding limited specific transcriptional change in the PCM1 nulls but a functionally suggestive set of alterations in the proteome that clearly implicate the satellites in centriole/ basal body/ ciliary protein homeostasis. However, these impacts are broad and an underlying pattern to the effects of satellite loss remains to be clarified.

The data presented here are strong and well-controlled. The revision of the manuscript has addressed the majority of the issues raised by the referees and notably improved the study. The refocussing of the paper has made the principal insights of the study much more evident to the reader. There are several areas where the information presented here will be of value to the further understanding of satellite functions and I am enthusiastic about this submission.

There remain some points that should be clarified, as follow:

Specific points

1. Does serum starvation impact the relative total levels of the proteins analysed in Fig. 4G? The authors have partially addressed this point, which was raised previously, in their immunofluorescence experiments, but the important question of targeting vs. total levels of the proteins of interest is not wholly resolved by the data here. (If this experiment was done on serum starved samples, this should be stated).
2. Does HU induce centriole overduplication to the same extent in the PCM1 null cells as in wild-type controls?

Minor points

3. Introduction: Centrosomes do not duplicate during mitosis and this should be corrected.
4. The Fig. 1 legend should specify what the asterisked protein is. The Fig. 1D legend should be corrected to say 1-1200 (no 3600).
5. The FACS plots in Fig. S2 could be more clearly presented, as in Supp. Fig 3G; Fig. S2's data are highly compressed.
6. It should be stated whether the IF data in Figure 4A-F were derived from all cells or those with cilia.
7. The asynchronous quantitation of Ift88 should be shown in Figure S4F. It would be helpful to have the Mib1, Talpid3 and Ift88 analyses in the same relative positions in Figs. 4 and S4 (i.e., swap D, E and F around).
8. The Ift88 'Ciliary Levels' data in Fig 5A appear to be the same as the 'Ciliary Ift88 Concentration' in Figure S5. This should be corrected, presumably by inserting the correct plot for the Ift88 'Ciliary Levels'.
9. Statistical non-significance (assuming this is the case) should be shown in the bar chart in Figure 6A to support the statement regarding the control and KO cells having similar percentages of Smo-positive cilia after 12 and 24h stimulation.
10. It would be helpful to include the protein names in Table S3 (the Table currently presents the gene names, which is not indicated in the Table legend).

Referee #2:

The authors have done a really good job in revising their manuscript and addressing the points that I have raised. This is a serious revision and I support publication in EMBO Reports without the need for further experiments (at least pertaining to my own comments on the original manuscript). There are still a couple of points that I don't feel have been addressed experimentally, as outlined below. However, the authors do make a good effort in discussing these issues, and the overall impression is that the data are very coherent.

- 1) The authors do dodge my comment on antibody specificity by citing other sources. It would have been nice to see that in their specific cell background and with their IF protocol, the antibody signals are specific (using e.g. siRNA).
- 2) They rescue the PCM1-associated defects in spheroid culture in figure 7 with a transient transfection of Myc-tagged PCM1. However, in the absence of co-staining for Myc it is impossible to assess whether these cells actually express the rescue construct.

Referee #1:

The key finding is the definition of the cilium-specific impacts of centriolar satellite ablation. This work is significant and extends our understanding of the roles of centriolar satellites and the mechanisms of ciliogenesis. This will be of general interest to the molecular biology community because centriolar satellites are poorly understood but of importance in a range of cellular activities. The principal finding of the study is robustly documented, using a range of different approaches.

Odabasi et al. here explore in detail the role of centriolar satellites in ciliary assembly and functioning by ablation of the key scaffold protein, PCM1, in murine IMCD3 kidney epithelial cells by CRISPR-mediated genome editing.

They demonstrate the loss of centriolar satellites resulting from PCM1 loss and a marked reduction of ciliogenesis in serum starved IMCD3 cells. A parallel experiment in human hTERT-RPE1 retinal pigmented epithelium cells showed an almost complete loss of ciliogenesis capacity, consistent with previously-published data in this cell line. They show no major impact of PCM1 ablation on cell cycle or centriole duplication. However, the authors show clearly that PCM1 loss affects the levels of key ciliary regulators in both serum starved and asynchronous cells. The intraflagellar transport complex B protein Ift88 levels declined at the basal body, whereas those of the appendage protein Cep164 increased. Notably, the basal body levels of Mib1 and Talpid3 were affected in a quantitatively different manner by PCM1 loss in IMCD3 cells compared with hTERT-RPE1 cells. Defective recruitment of ciliary proteins Ift88, Sstr3 and Htr6 was seen in the absence of PCM1 and a clear deficiency in the movement to cilia of Smo and Hh activation was also observed. In a 3D spheroid model of kidney epithelial cell assembly, PCM-deficient cells were notably defective and also showed a decline in ciliation. The authors then present a detailed analysis of the impact of PCM1 deficiency on the IMCD3 transcriptome and proteome, finding limited specific transcriptional change in the PCM1 nulls but a functionally suggestive set of alterations in the proteome that clearly implicate the satellites in centriole/ basal body/ ciliary protein homeostasis. However, these impacts are broad and an underlying pattern to the effects of satellite loss remains to be clarified.

The data presented here are strong and well-controlled. The revision of the manuscript has addressed the majority of the issues raised by the referees and notably improved the study. The refocussing of the paper has made the principal insights of the study much more evident to the reader. There are several areas where the information presented here will be of value to the further understanding of satellite functions and I am enthusiastic about this submission.

We thank the reviewer for the accurate summary of our findings and for the positive comments about our work. We are very happy to see the reviewer found the data presented in the manuscript as robust and well described.

There remain some points that should be clarified, as follow:

Specific points

1. Does serum starvation impact the relative total levels of the proteins analysed in Fig. 4G? The authors have partially addressed this point, which was raised previously, in their immunofluorescence experiments, but the important question of targeting vs. total levels of the proteins of interest is not wholly resolved by the data here. (If this experiment was done on serum starved samples, this should be stated).

We thank the reviewer for pointing this out. In fact, the comparison of total cellular abundance of proteins between control and PCM1 KO cells in Fig. 4G was done using extracts prepared from an asynchronous population. Therefore, we now moved these figures to Fig. EV4G and clarified the conditions in Fig. EV4 legends. To correlate the changes in cellular abundance of proteins with the changes in their centrosomal abundance, we now included immunoblotting data of lysates prepared from cells serum starved for 24 h in Fig. 4G and clarified the conditions in Fig. 4 legends.

2. Does HU induce centriole overduplication to the same extent in the PCM1 null cells as in wild-type controls?

I agree with the reviewer that it will be interesting to determine whether PCM1 functions in centriole overduplication. However, given that IMCD3 and RPE1 cells do not overduplicate their centrioles after HU treatment, we will not be able to perform these experiments in the cell lines we used in this study. Based on literature, p53-deficient cells, in particular, U2OS and CHO cells, overduplicate their centrioles when arrested in S phase (HU treatment) (Tarapore, P et al. Oncogene PMID: 12214254, Balczon, R et al. JCB PMID: 2120504).

Minor points

3. Introduction: Centrosomes do not duplicate during mitosis and this should be corrected.

We replaced this sentence with “Centrosomes duplicate during S phase and form the bipolar mitotic spindle during mitosis”.

4. The Fig. 1 legend should specify what the asterisked protein is. The Fig. 1D legend should be corrected to say 1-1200 (no 3600).

We corrected this as suggested.

5. The FACS plots in Fig. S2 could be more clearly presented, as in Supp. Fig 3G; Fig. S2's data are highly compressed.

We have now rescaled the FACS plots and labeled them for cell cycle phases for clearer presentation.

6. It should be stated whether the IF data in Figure 4A-F were derived from all cells or those with cilia.

We now included in Fig.4 and Fig. EV4 legends the statement “Both ciliated and unciliated cells were quantified in a blinded manner.” Of note, we also compared the levels between ciliated and unciliated cells and did observe similar phenotypes to the ones we report, therefore we did not include that data in the manuscript.

7. The asynchronous quantitation of Ift88 should be shown in Figure S4F. It would be helpful to have the Mib1, Talpid3 and Ift88 analyses in the same relative positions in Figs. 4 and S4 (i.e., swap D, E and F around).

We now reorganized Fig. 4 as suggested and included quantification of Ift88 centrosomal levels in asynchronous cells in Fig. EV4D.

8. The Ift88 'Ciliary Levels' data in Fig 5A appear to be the same as the 'Ciliary Ift88 Concentration' in Figure S5. This should be corrected, presumably by inserting the correct plot for the Ift88 'Ciliary Levels'.

As the reviewer suggested, we mislabeled Fig. S5 as “Ciliary Ift88 concentration”. We now corrected it to “Ciliary Ift88 levels”.

9. Statistical non-significance (assuming this is the case) should be shown in the bar chart in Figure 6A to support the statement regarding the control and KO cells having similar percentages of Smo-positive cilia after 12 and 24h stimulation.

We included the statistical non-significance label in Fig. 6A for 12h and 24h SAG stimulation datapoints.

10. It would be helpful to include the protein names in Table S3 (the Table currently presents the gene names, which is not indicated in the Table legend).

We now included the protein names and their description in Table S3 and revised the Table legend was modified to include the following sentence: “The table includes gene names, protein IDs and descriptions”.

Referee #2:

The authors have done a really good job in revising their manuscript and addressing the points that I have raised. This is a serious revision and I support publication in EMBO Reports without the need for further experiments (at least pertaining to my own comments on the original manuscript). There are still a couple of points that I don't feel have been addressed experimentally, as outlined below. However, the authors do make

a good effort in discussing these issues, and the overall impression is that the data are very coherent.

We thank the reviewer for acknowledging the work we put in to the first revision of our manuscript and for supporting publication in EMBO Reports.

1) The authors do dodge my comment on antibody specificity by citing other sources. It would have been nice to see that in their specific cell background and with their IF protocol, the antibody signals are specific (using e.g. siRNA).

- We are sorry that our rebuttal to this comment about specificity came out as ignorance. In fact, as suggested by the reviewer in the same comment as a way to test specificity of phenotypes for changes in protein levels, we performed quantifications for all proteins we reported changes in the rescue cell line. Given that we observed significant rescue in all cases, namely IFT88, Mib1, Talpid3 and Cep164, we concluded that the phenotypes we reported are specific to PCM1 KO.

- Regarding testing the antibody specificities by siRNA experiments, I would like to note that we quantified a total of 8 centrosomal and ciliary proteins by immunofluorescence and we do not have siRNAs available for all of them in the lab, except for Cep290. We validated Cep290 antibody specificity with its siRNA and can include this data in the supplement. However, hopefully without offending the reviewer, I would like to highlight again that all these antibodies were used and validated previous studies. In addition to antibody RRIDs, we now included citations of the papers that used and validated these antibodies in the materials and methods. Moreover, the key ciliogenesis factors we tested in our study and the antibodies against them are frequently also used for immunofluorescence in papers related to mechanistic dissection of ciliogenesis pathways as exemplified by Wang L et al. eLife 2016 PMID: 27146717, Agbu SO et al. 2018 JCB PMID: 5748968, Tu et al. Nature Communications 2018 PMID: 6290075. Since none of the proteins we quantified are novel proteins, I believe that rescue experiments and citing the papers that has validation information for the antibodies we used will be sufficient to address the reviewer's concern on specificity.

2) They rescue the PCM1-associated defects in spheroid culture in figure 7 with a transient transfection of Myc-tagged PCM1. However, in the absence of co-staining for Myc it is impossible to assess whether these cells actually express the rescue construct.

-When we stained the spheroid cultures with PCM1 antibodies that work for immunofluorescence in 2D cultures, we did not see PCM1-positive staining even in wild type cells. Either due to antibody accessibility issues or changes in the distribution of satellites in 3D cultures, we were not able to visualize PCM1 signal in 3D cultures.

- To confirm that myc-PCM1 is expressed in cells used for spheroid rescue experiments, we cultured IMCD3 cells transfected with myc-PCM1 in 2D in parallel to 3D culturing and validated myc-PCM1 expression in 2D cultures using experiments. Fig. EV7 includes fluorescent micrographs of cells stained for myc, PCM1 and gamma-

tubulin, confirming the expression of myc-PCM1. Likely due to the presence of untransfected cells in the population, we do only see a partial rescue in spheroid formation efficiency (Fig. 7A).

Accepted

1st April 2019

Thank you for submitting your revised manuscript to EMBO Reports. I have now looked at everything and all looks fine. Therefore I am very pleased to accept your manuscript for publication in EMBO Reports.

Corresponding Author Name: Elif Nur Firat-Karalar

Manuscript Number: EMBOR-2019-47723